# Stable MOB1 interaction with Hippo/MST is not essential for development and tissue growth control

Yavuz Kulaberoglu [1,6], Kui Lin[2], Maxine Holder[3], Zhongchao Gai[2], Marta Gomez[1], Belul Assefa Shifa[1], Merdiye Mavis[1], Lily Hoa[1], Ahmad A.D. Sharif[1], Celia Lujan[4], Ewan St. John Smith [5], Ivana Bjedov[4], Nicolas Tapon[3], Geng Wu[2] & Alexander Hergovich[1]

The Hippo tumor suppressor pathway is essential for development and tissue growth control, encompassing a core cassette consisting of the Hippo (MST1/2), Warts (LATS1/2), and Tricornered (NDR1/2) kinases together with MOB1 as an important signaling adaptor. However, it remains unclear which regulatory interactions between MOB1 and the different Hippo core kinases coordinate development, tissue growth, and tumor suppression. Here, we report the crystal structure of the MOB1/NDR2 complex and define key MOB1 residues mediating MOB1's differential binding to Hippo core kinases, thereby establishing MOB1 variants with selective loss-of-interaction. By studying these variants in human cancer cells and *Drosophila*, we uncovered that MOB1/Warts binding is essential for tumor suppression, tissue growth control, and development, while stable MOB1/Hippo binding is dispensable and MOB1/Trc binding alone is insufficient. Collectively, we decrypt molecularly, cell biologically, and genetically the importance of the diverse interactions of Hippo core kinases with the pivotal MOB1 signal transducer.

[1] Tumour Suppressor Signalling Network Laboratory, UCL Cancer Institute, University College London, London WC1E 6BT, UK. [2] State Key Laboratory of Microbial Metabolism, School of Life Sciences & Biotechnology, Shanghai Jiao Tong University, Shanghai 200240, China. [3] Apoptosis and Proliferation Control Laboratory, Francis Crick Institute, London NW1 1BF, UK. [4] Molecular Biology of Cancer laboratory, UCL Cancer Institute, University College London, London WC1E 6BT, UK. [5] Department of Pharmacology, University of Cambridge, Cambridge CB2 1PD, UK. [6] Department of Pharmacology, University of Cambridge, Cambridge CB2 1PD, UK. Yavuz Kulaberoglu and Kui Lin contributed equally to this work. Correspondence and requests for materials should be addressed to Department.of.Pharmacology, University.of.Cambridge, Cambridge CB2 1PDUKG.W. (email: geng.wu@sjtu.edu.cn) or to A.H. (email: a.hergovich@ucl.ac.uk)

The Hippo pathway is vital for organ growth control and tissue homeostasis[1–3], and its dysregulation has been linked to various human cancers[4]. Therefore, it is imperative to understand how major core components of the Hippo pathway are regulated in different cellular contexts in health and disease. The Hippo tumor suppressor pathway signals through three main levels: (i) a central core kinase cassette, (ii) downstream regulators of transcriptional programs, and (iii) diverse upstream regulators[1–4]. The *Drosophila* core cassette comprises the Warts (Wts) and Hippo (Hpo) kinases supported by the adaptor protein Mats[5, 6]. The Hpo-Wts-Mats cassette acts together to inhibit Yorkie (Yki), which when uncontrolled promotes tissue overgrowth[7]. The Hippo core cassette is conserved from flies to humans, with human MST2, LATS1, and MOB1A compensating for Hpo, Wts, and Mats loss-of-function, respectively[8–10]. Thus, *Drosophila* is well suited for in vivo studies of essential molecular mechanisms in Hippo core signaling[5, 11].

The mammalian Hippo core cassette contains MST1/2, LATS1/2, and MOB1 which together regulate the transcriptional co-activators YAP/TAZ[12, 13]. MST1/2, LATS1/2, MOB1, and YAP/TAZ are the human equivalents of fly Hpo, Wts, Mats, and Yki[7, 14–16]. Upon activation MST1/2 phosphorylates LATS1/2 and MOB1 to support formation of an active MOB1/LATS complex, which phosphorylates YAP/TAZ, promoting its cytoplasmic retention and/or degradation[3]. The NDR1/2 kinases, the closest relatives of LATS1/2[15], are also controlled by MST1/2 and MOB1[17], and regulate YAP[18]. Thus, the Hippo core pathway can act through distinct kinases, with MOB1 acting as a fundamental signal adaptor that can interact with the Hippo core kinases MST1/2, LATS1/2, and NDR1/2[16, 19–22].

MOB proteins are highly conserved amongst eukaryotes, constituting signal transducers in essential processes via their regulatory interactions with NDR/LATS[16]. In *Drosophila*, Mats (aka dMOB1[16]) can function together with Wts and Hpo in Hippo signaling[10, 23]. However, Mats also interacts genetically with Trc[24], the fly counterpart of human NDR1/2[15]. Mammals express at least six MOBs, with MOB1A and MOB1B sharing 95% sequence identity[16]. MOB1A/B (aka MOB1) can function redundantly as regulators of LATS1/2 signaling[21, 25–27], but are also required for NDR1/2 activation[16], as an alternative branch[17, 18] of the Hippo core cassette. Biochemical evidence suggests that MOB1 can associate with the highly conserved N-terminal regulatory domain (NTR) of NDR/LATS kinases to promote their activities[15, 16, 25]. MST1/2 phosphorylation of MOB1 can influence MOB1 binding to the NTR[19, 20, 22, 28, 29], although it is yet to be determined whether this phosphorylation is required for NDR/LATS kinase activity in vivo. Nevertheless, MOB1 (Mats) is very likely acting as a central molecular switch in Hippo signaling[3, 30]. However, we still lack structural and molecular insights on how the regulatory bindings of MOB1 to MST1/2, LATS1/2, or NDR1/2 are mediated and how MOB1 differentiates between these interactions. Two crystal structures of MOB1 bound to LATS1 were reported[19, 20], but the crystal structure of the MOB1/NDR complex has yet to be documented. The importance of stable MOB1 binding to MST1/2 (Hpo) is currently also debatable based on recently published biochemical data and the analysis of a chimeric conformation sensor[19, 20, 28–31]. In general, the biological significance of MOB1 interactions with Hippo core kinases is not defined for tumor suppression, development, and tissue growth control (Supplementary Fig. 1a).

Here, we report our structural, biochemical, cell biological, and organismal studies of the importance of MOB1 interactions with Hippo core kinases. The overall crystal structure of MOB1 bound to NDR2 is very similar to the previously reported structures of MOB1/LATS1[19, 20]. However, by comparing these crystal structures we could identify Asp63 as a key MOB1 residue that specifically mediates binding to LATS1. Thus, we characterized the interactions of Hippo core kinases with full-length MOB1 variants carrying specific point mutations, resulting in the discovery of MOB1 variants that are selectively impaired in their binding to MST1/2 (Hpo) or LATS1/2 (Wts) in human and fly cells. Using these MOB1 variants with selective loss-of-interaction, we found that a stable interaction of MOB1 with LATS1/2, but not with MST1/2, is essential for tumor suppressive properties of MOB1 in human cancer cells. By employing fly genetics, we discovered that the MOB1/Wts interaction is essential for development and tissue growth control, while stable MOB1 binding to Hpo is dispensable. Taken together, our study decrypts the nature and functional importance of the diverse interactions of Hippo core kinases with the central MOB1 signaling adaptor.

## Results

### Crystal structure of MOB1 bound to the NTR of human NDR2.

To delineate the interaction of MOB1 with NDR2 on the atomic level, we determined the crystal structure of the MOB1/NDR2 complex at 2.1 Å using purified MOB1 (residues 33–216) and the NTR of NDR2 (residues 25–88) (Fig. 1a–c and Table 1). The structure of MOB1 adopts a globular shape consisting of nine α-helices (α1–α9) and two β-strands (Fig. 1a), as reported for unbound human MOB1[19, 32], frog MOB1[33], and yeast Mob1p[34]. The overall structure of the MOB1/NDR2 complex is similar to the reported MOB1/LATS1 complex[19, 20], since NDR2 binds to MOB1 in a V-shaped structure composed of two antiparallel α-helices (Fig. 1a–c). In agreement with biochemical studies[15, 17, 25], highly conserved positively charged residues of NDR2 bond with negatively charged electrostatic surfaces of MOB1 (Fig. 1b, c).

The central intermolecular interactions between MOB1 and NDR2 are hydrogen bonds and van der Waals interactions (Fig. 1d), with the two antiparallel α-helices of NDR2 representing two interaction interfaces (Fig. 1e, f). In the α1 helix Lys25, Leu28, Tyr32, Leu35, and Ile36 of NDR2 interact with Leu36, Gly39, Leu41, Ala44, Gln67, Met70, Leu71, Leu173, Gln174, and His185 of MOB1 (Figs. 1d, e and 2a). In the α2 helix Arg42, Leu78, Arg79, and Arg82 of NDR2 interact with Glu51, Glu55, Trp56, Val59, Phe132, Pro133, Lys135, and Val138 of MOB1 (Figs. 1d, f and 2a). The V-shape of the bihelical NTR domain of NDR2 is stabilized by intramolecular interactions of Arg45 and Glu50 in the α1 helix with Arg67 and Glu74 in the α2 helix (Fig. 1f).

Our structural data are supported by previous biochemical studies of NDR1 (NDR2) variants carrying single point mutations at K24A (K25A), Y31A (Y32A), R41A (R42A), R44A (R45A), T74A (T75A), or R78A (R79A) in the context of MOB1 binding and NDR1/2 activation[15, 17, 25]. Biochemical studies of LATS1/2 mutants carrying substitutions of the residues corresponding to Arg42, Arg45, Glu74, Thr75, Arg79, or Arg82 of NDR2 also demonstrated the importance of these conserved residues for MOB1 binding and LATS1/2 activation[19–21, 35, 36].

### MOB1 binds differently to the NTRs of NDR2 and LATS1 kinases.

To define possible differences between MOB1/NDR2 and MOB1/LATS1 complexes, we compared available MOB1/LATS1 structures[19, 20] with our MOB1/NDR2 structure (Fig. 1). This revealed fully conserved core interactions, but also dissimilarities (Fig. 2b and Supplementary Fig. 2). Most significantly, we discovered that His646 of LATS1 bonds with Asp63 of MOB1 (Fig. 2b) supported by a cluster of surrounding residues involving Phe642, Met643, Gln645, Val647, and Val650[19], while Phe31 of NDR2 does not interact with Asp63 of MOB1 (Figs. 1d, e and 2a,

b). Consequently, our structural comparison suggests that Asp63 of MOB1 specifically bonds with LATS kinases through His646 (Fig. 2b), which is conserved in human LATS1/2 and fly Wts, but replaced by a bulky Phe/Tyr residue in human NDR1/2 and fly Trc (Fig. 2a). Thus, our evidence indicates that MOB1 binds differently to NDR vs. LATS kinases.

To investigate whether the interaction thermodynamics differ, we performed isothermal titration calorimetry (ITC) assays to determine the dissociation constant ($K_d$) of full-length MOB1 with the NTRs of NDR1, NDR2, LATS1, or LATS2 (Fig. 2c–f and Supplementary Fig. 3). Significantly, unphosphorylated full-length MOB1 bound to NDR1 and NDR2 (Fig. 2c, *left panel* and Supplementary Fig. 3a), while an interaction with LATS1 or LATS2 was undetectable (Fig. 2d, *left panel*, and Supplementary Fig. 3b). MOB1(Q67A) and MOB1(H185A) mutants did not interact with NDR2 (Supplementary Fig. 4), illustrating that the

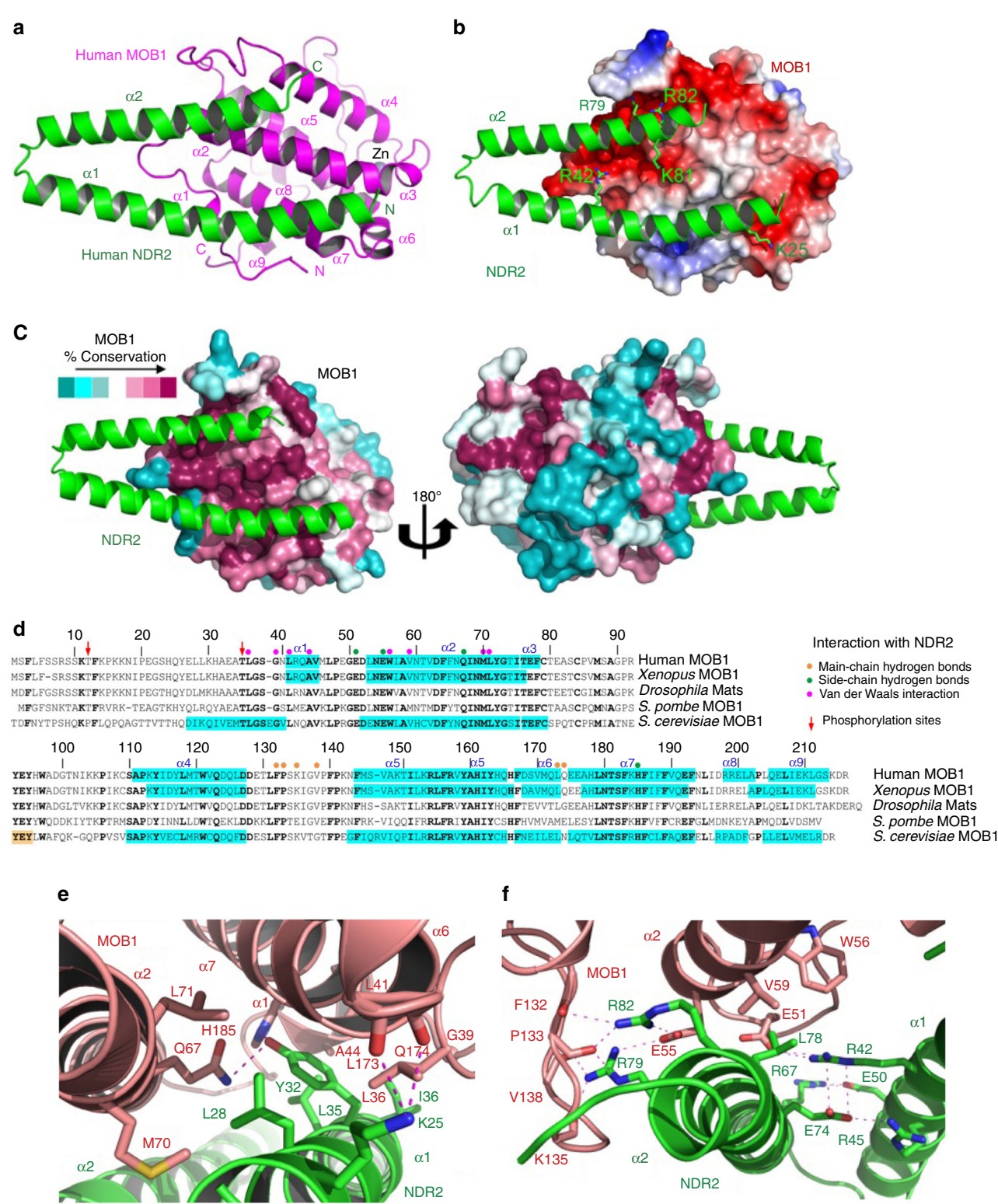

observed MOB1/NDR2 interaction is specific. Considering that MST1/2 phosphorylation of MOB1 can influence *in vitro* MOB1 binding to the NTR of LATS1[19, 20, 22, 28, 29], we measured the $K_d$ of MST1/2-phosphorylated full-length MOB1 (phospho-MOB1) with NDR1, NDR2, LATS1, or LATS2 (Fig. 2c, d, *right panels*, and Supplementary Fig. 3a, b). This revealed that MOB1 phosphorylation is essential for MOB1 binding to LATS1 or LATS2 in our experimental settings using protein fragments representing the NTR domains (Fig. 2d and Supplementary Fig. 3b). We further uncovered that NDR1 and NDR2 bound with enhanced affinity to phospho-MOB1 than to non-phosphorylated MOB1 (Fig. 2c and Supplementary Fig. 3a). Phospho-MOB1 also bound with an at least 15-fold (or higher) increased affinity to NDR1 or NDR2 compared to LATS1 or LATS2 (Fig. 2c, d, *right panels* and Supplementary Fig. 3a, b).

Since we discovered Asp63 of MOB1 as a potential key determinant in the differential binding of MOB1 to NDR2 vs. LATS1 (Fig. 2b), we tested the contribution of residues surrounding His646 of LATS1 and Phe31 of NDR2 to MOB1 binding. Specifically, considering that Tyr32 of NDR2 (and Tyr31 of NDR1, respectively) is essential for MOB1 binding and kinase activation[15, 17, 25], but not conserved in LATS1/2[15, 17, 25], we hypothesized that by switching Val647 of LATS1 (or Val610 of LATS2) to a tyrosine we may change the binding affinities of LATS1 and LATS2 into the ones observed for NDR1 and NDR2. Thus, we studied NDR1(Y31V), NDR2(Y32V), LATS1(V647Y), and LATS2(V610Y) mutants. As expected based on the central role of Tyr32 in MOB1/NDR2 complex formation (Figs. 1e and 2a), NDR2(Y32V) and NDR1(Y31V) did not associate with non-phosphorylated MOB1 (Fig. 2e, *left panel*, and Supplementary Fig. 3c). However, NDR2(Y32V) and NDR1(Y31V) bound to phospho-MOB1 (Fig. 2e, *right panel*, and Supplementary Fig. 3c), although with a 20-fold decreased binding affinity compared to wild-type NDR2 or NDR1, respectively (compare Fig. 2c, e and Supplementary Fig. 3a, c). Significantly, LATS1(V647Y) and LATS2(V610Y) interacted with non-phosphorylated MOB1 with a similar $K_d$ as observed between wild-type LATS1 or LATS2 and phospho-MOB1 (compare Fig. 2d, f and Supplementary Fig. 3b, d). LATS1(V647Y) and LATS2(V610Y) even displayed a 5-fold increased binding affinity for phospho-MOB1 compared to wild-type LATS1 or LATS2, respectively (compare Fig. 2d, f, and Supplementary Fig. 3b, d).

Taken together, we demonstrate in Figs. 1 and 2 that MOB1 relies on different residues to bind to NDR2 and LATS1. MST1/2 phosphorylation of MOB1 can play a significant role in modulating in vitro the diverse binding affinities of MOB1 to the NTRs of NDR/LATS kinases. MOB1/NDR2 complex formation is dramatically increased by prior MST1/2 phosphorylation of MOB1, while the MOB1/LATS1 interaction appears to be fully dependent on MST1/2 phosphorylation of MOB1. In this regard, a single substitution of Val647 to Tyr of LATS1 (or Val610 of LATS2) is sufficient to support MOB1/LATS complex formation independent of MOB1 phosphorylation. More specifically, LATS1(V647Y) and LATS2(V610Y) bound to non-phospho and phospho-MOB1 with much increased affinities as observed for LATS1 and LATS2 wild-type, hence indicating that a single substitution in LATS1 or LATS2 can switch the binding mode of LATS kinases. Noteworthy, these conclusions are solely based on our ITC assays (Fig. 2 and Supplementary Fig. 3), hence the in vivo relevance has yet to be determined. In this regard, it has been documented that unphosphorylated MOB1 can bind to a LATS1 fragment in vitro[19, 20]. Therefore, more research is needed to decipher the reason(s) for this discrepancy. Possibly our ITC assay requires higher concentrations to detect lower affinity interactions, or ITC assays with higher affinity kinase fragments are necessary. Certainly, identical kinase fragments will need to be tested to allow a proper comparison of our findings (Fig. 2 and Supplementary Fig. 3) with previous studies[19, 20], but most importantly, the in vivo implications of MST1/2 (Hpo) phosphorylation of MOB1 need to be delineated in future studies.

### Table 1 Data collection and refinement statistics

| Data collection | |
| --- | --- |
| Space group | $P2_12_12_1$ |
| Unit-cell parameters | $a = 57.5$ Å, $b = 94.4$ Å, $c = 102.2$ Å; $\alpha = \beta = \gamma = 90°$ |
| Number of molecules/asymmetric unit | 2 |
| Resolution range (Å) | 50-2.10 (2.18-2.10) |
| Completeness (%) | 99.6 (100.0) |
| Redundancy | 6.9 (6.4) |
| Total observations | 226,859 |
| Unique reflections | 32,723 |
| $R_{merge}$ (%) | 10.0 (61.3) |
| $I/\sigma_I$ | 16.3 (3.1) |
| Refinement | |
| $R_{work}$ (%) | 15.3 |
| $R_{free}$ (%) | 24.4 |
| Overall B factor | 42.3 |
| RMSD bond lengths (Å) | 0.013 |
| RMSD bond angles (°) | 1.507 |
| Ramanchandran plot (favored, allowed, disallowed, %) | 99.0, 0.6, 0.4 |
| Final model (number of protein/solvent atoms) | 4,149/267 |

*RMSD* root-mean-square deviations from ideal geometry. $R_{merge} = \Sigma_h \Sigma_i |I_{h,i} - I_h| / \Sigma_h \Sigma_i I_{h,i}$ for the intensity (*I*) of observation i of reflection h. R factor = $\Sigma ||F_{obs}| - |F_{calc}|| / \Sigma |F_{obs}|$, where $F_{obs}$ and $F_{calc}$ are the observed and calculated structure factors, respectively. $R_{free} = R$ factor calculated using 5% of the reflection data chosen randomly and omitted from the start of refinement. Data for the highest resolution shell are shown in parentheses

**Fig. 1** The crystal structure of the MOB1/NDR2 complex reveals phosphorylation-independent interactions of MOB1 with NDR2 through conserved interfaces. **a** Overall crystal structure of the complex between human MOB1A (residues 33–216) and human NDR2 (residues 25–88). MOB1 and NDR2 are colored in *magenta* and *green*, respectively. The Zinc associated with MOB1 is indicated in *gray*. Secondary structure elements are highlighted. **b** The negatively charged surface of MOB1 recognizes positively charged residues on NDR2. The electrostatic surface potential of MOB1 is shown (*red*: negatively charged residues; *blue*: positively charged residues). NDR2 is shown as a ribbon representation colored in *green*. **c** The MOB1/NDR2 binding surfaces are highly conserved. The residues of MOB1 are gradually colored according to conservation scores (*dark red*: most conserved residues; *dark cyan*: least conserved residues). NDR2 is shown as a ribbon representation colored in *green*. **d** Structure based sequence alignment of MOB1 from indicated species. MOB1 residues mediating interactions with NDR2 are marked with *brown* (main-chain hydrogen bonds), *green* (side-chain hydrogen bonds), and *magenta circles* (van der Waals interactions). α helices are painted with *cyan* background. Thr12 and Thr35 phosphorylation sites are indicated by *red arrows*. **e, f** The interaction interfaces of MOB1 with the α1 (residues 25–56) and α2 (residues 61–84) helices of NDR2. MOB1 and NDR2 and their key interacting residues are shown in *pink* and *green*, respectively. *Dashed lines* represent hydrogen bonds. Side-chain nitrogen, oxygen, and sulfur atoms of MOB1 and NDR2 residues are colored in *blue*, *red* and *gold*, respectively. The hydrogen bond network connecting the α1 and α2 helices of NDR2 is also illustrated in **f**. For a stereo image of a portion of the electron density map see Supplementary Fig. 18

**Defining MOB1 variants with selective loss-of-interactions.** To empower the translation of our structural and biochemical findings into studies of human cells and flies, we studied the interactions of full-length human MOB1 variants with human/fly Hippo core kinases (Fig. 3 and Supplementary Figs. 5–7). In human HEK293 cells myc-tagged MOB1 mutants were co-expressed with HA-tagged wild-type kinases, followed by co-immunoprecipitation experiments in low-stringency conditions (Supplementary Figs. 5–7). Moreover, myc-tagged MOB1 binding to fly HA-tagged Wts, Trc, and Hpo was examined in *Drosophila* S2R + cells (Supplementary Figs. 5e, 6e, f, and 7e). These experiments revealed that MOB1(D63V) does not form stable complexes with LATS1/2 or Wts, while stably associating with NDR1/2, Trc, MST1/2, and Hpo (Supplementary Figs. 5–7).

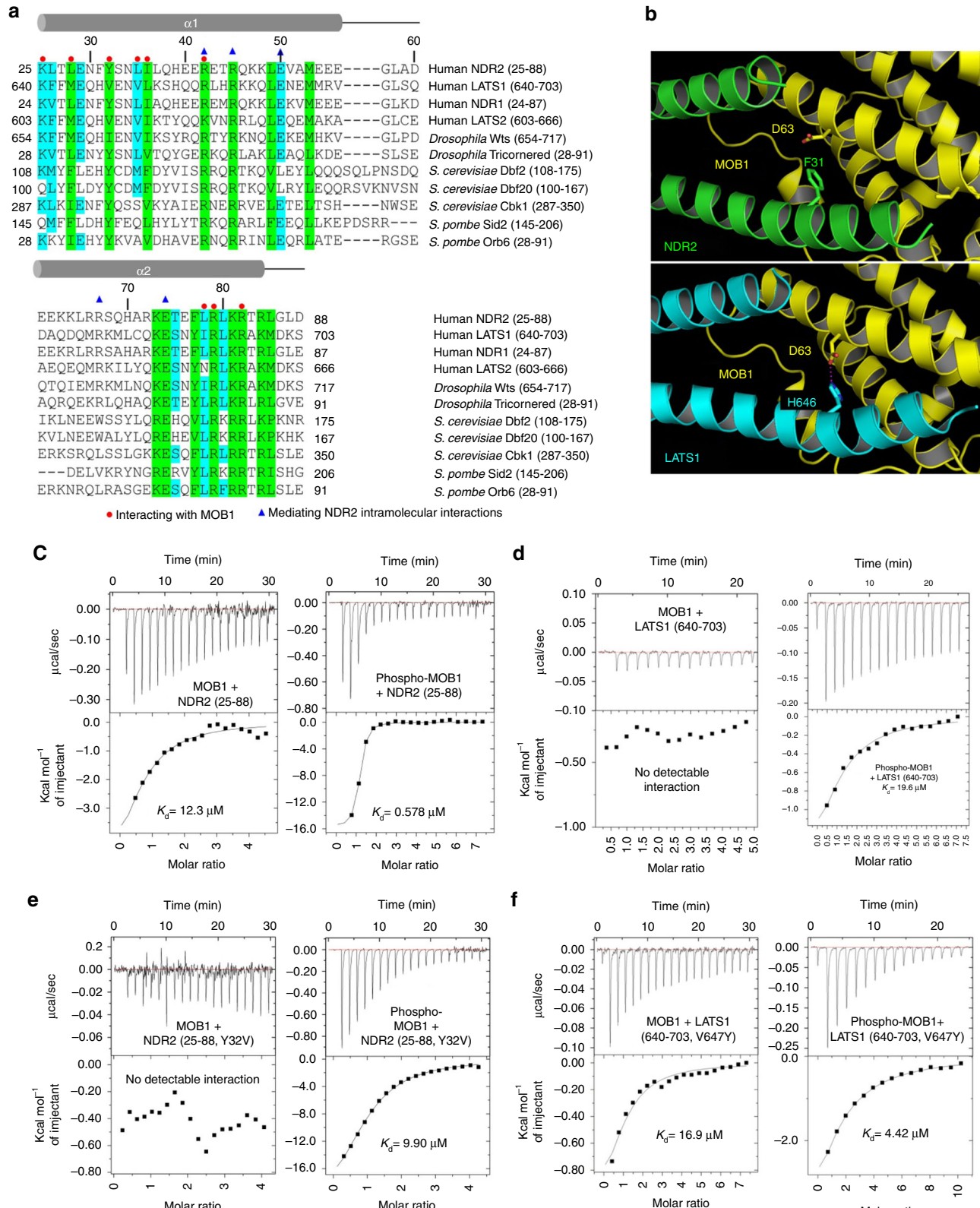

Thus, together with our structural data (Figs. 1 and 2) these findings indicate that Asp63 of MOB1 plays a specific role in MOB1 binding to fly and human LATS kinases.

To study the entire spectrum of MOB1 interactions with Hippo core kinases (Supplementary Fig. 1a), we next determined key residues mediating MOB1 binding to MST1/2 and Hpo. As observed for the MOB1 interactions with NDR/LATS kinases (Figs. 1 and 2 and refs. [19, 20]), we speculated that charged residues of MOB1 are also centrally important for stable complex formation with human MST1/2 and fly Hpo. Consequently, we performed co-immunoprecipitation assays with MST1/2 and a panel of MOB1 variants. Significantly, a MOB1(K104E/K105E) version failed to stably bind to MST1/2 and Hpo (Fig. 3a, b). MOB1(K104E/K105E) displayed selective loss-of-interaction with MST1/2 and Hpo since it was still proficient in binding to LATS1/2, Wts, NDR1/2 or Trc (Fig. 3a, b and Supplementary Figs. 5–7). Moreover, unlike wild-type MOB1, MOB1(K104E/K105E) did not associate with human MST2 in three additional co-immunoprecipitation conditions (Supplementary Fig. 8). Collectively, these data suggest that Lys104 and Lys105 of MOB1 play specific roles in MOB1 binding to Hpo and MST1/2.

In the hope to generate an alternative MOB1 variant displaying selective loss-of-interaction with MST1/2 and Hpo, we considered modifications of the MOB1 residues Lys153 and Arg154 as promising candidates due to their central roles in the phospho-threonine binding interface supporting MOB1/MST1 and MOB1/MST2 interactions[20, 28]. However, MOB1(K153A/R154A) and MOB1(K153E/R154E) displayed defective interactions with full-length wild-type MST2 and LATS2, while NDR1 binding was intact (Supplementary Fig. 9). MOB1(K153A/R154A) and MOB1(K153E/R154E) were also defective in Wts and Hpo binding, while they bound to Trc in insect cells (Supplementary Fig. 9). Thus, as suggested previously[37], as central P0 phosphate coordinating residues[28, 37], Lys153 and Arg154 are likely to represent the core of a more general phospho-serine/threonine binding domain of MOB1. This conclusion is reinforced by the finding that the region surrounding Lys153 and Arg154 of MOB1 also supports Praja2 binding[38]. As a result, we concluded that Lys153 and Arg154 modifications of MOB1 are not suitable to develop MOB1 variants with selective loss-of-interaction with MST1/2 and Hippo.

Since MOB1(E51K) is deficient in NDR1/2 binding[39], we also profiled MOB1(E51K) (Supplementary Figs. 5–7). However, as summarized in Fig. 3c, MOB1(E51K) associated inconsistently with Hippo core kinases and was therefore excluded from further cellular studies. Alternatively, we engineered a MOB1(D63V/K104E/K105E) mutant to establish a MOB1 version that only associates with NDR1/2, but not with LATS1/2 and MST1/2 (Fig. 3c and Supplementary Figs. 5–7).

To complete the biochemical characterization, we performed additional experiments. First, we investigated the importance of Asp63 and Lys104/Lys105 of MOB1 for in vitro binding to Hippo core kinases using gel filtration chromatography, revealing that recombinant full-length MOB1(D63V) and MOB1(K104E/K104E) displayed the same binding patterns as observed for full-length proteins expressed in human and fly cells (Fig. 3c and Supplementary Fig. 10). Second, we performed ITC assays to determine the dissociation constant of full-length MOB1(K104E/K105E) with the NTRs of NDR1, NDR2, LATS1, and LATS2 (Supplementary Fig. 11). Unphosphorylated MOB1(K104E/K105E) bound to all four NTRs comparable to unphosphorylated wild-type MOB1 (compare Fig. 2 and Supplementary Figs. 3 and 11). Likewise, phospho-MOB1(K104E/K105E) displayed similar affinities to all four NTRs as observed for wild-type MOB1 (compare Fig. 2 and Supplementary Figs. 3 and 11). Binding of MOB1(K104E/K105E) to NDR1(Y31V), NDR2(Y32V), LATS1(V647Y), and LATS2(V610Y) NTR mutants was also comparable to wild-type MOB1 (compare Fig. 2 and Supplementary Figs. 3 and 11). Third, we measured MST1/2 (Hpo) phosphorylation of our MOB1 variants (Supplementary Figs. 12 and 13), since MOB1 phosphorylation can influence the binding affinities of MOB1 to NDR/LATS (Fig. 2 and refs. [19, 22, 28, 29]). MST1/2 (Hpo) phosphorylation of MOB1 on Thr12 and Thr35 was comparable for all MOB1 versions tested (Fig. 3c and Supplementary Figs. 12 and 13). Thus, the selective loss-of-interaction of MOB1(D63V) is not a consequence of altered Thr12/Thr35 phosphorylation, but rather caused by the substitution of a key residue that is essential for MOB1/LATS complex formation. Our data (Fig. 3 and Supplementary Figs. 7–13) further argue that a stable interaction of MST1/2 (Hpo) with MOB1 is not required for MOB1 phosphorylation by MST1/2 (Hpo).

Taken together, we discovered that distinct MOB1 residues mediate the interactions with the different mammalian and fly Hippo core kinases. Specifically, Asp63 and Lys104/Lys105 of MOB1 represent key residues mediating the differential binding properties of MOB1 with LATS1/2 (Wts) and MST1/2 (Hpo), respectively (Fig. 3d).

**Testing of MOB1 variants in anchorage-independent growth.** To define which MOB1 interactions with Hippo core kinases are necessary for tumor suppression, we engineered pools of MCF-7 human breast cancer cells stably expressing either empty vector (EV), HA-MOB1 wild-type (wt), HA-MOB1(D63V), HA-MOB1(K104E/K105E), or HA-MOB1(D63V/K104E/K105E) (Fig. 4a). Then, we determined proliferation and colony formation in two-dimensional (2D) culture conditions. Specifically, we measured proliferation using IncuCyte live cell analysis technology (Fig. 4b) and performed colony formation assays (Fig. 4c, d) to determine

**Fig. 2** MOB1 binds differently to the NTR domains of NDR2 and LATS1 through key residues. **a** Structure based sequence alignment of the conserved N-terminal regulatory domain of NDR/LATS kinases that is required for MOB1 binding. NDR2 residues mediating interactions with MOB1 are marked with *red circles*. Residues mediating intramolecular interactions of NDR2 are denoted by *blue triangles*. Residues conserved from yeast to humans are highlighted in *green*, while residues conserved in at least seven of the conserved NDR/LATS family members are marked in *cyan*. α helices are indicated with *gray rods*. **b** Asp63 of MOB1 is involved in LATS1 binding, but does not contribute to the interaction of MOB1 with NDR2. MOB1 is shown in *yellow*, while NDR2 and LATS1 are indicated in *green* and *cyan*, respectively. *Top panel*, Phe31 of NDR2 does not interact with Asp63 of MOB1. *Bottom panel*, His646 of LATS1 forms a hydrogen bond with Asp63 of MOB1. **c–f** Isothermal titration calorimetry (ITC) assays measuring the dissociation constant ($K_d$) of indicated non-phosphorylated full-length MOB1 or MST2-phosphorylated MOB1 (phospho-MOB1) with wild-type and mutant NDR2 (25–88) or LATS1 (640–703) variants. MOB1/NDR2 complex formation was dramatically increased by prior phosphorylation of MOB1 **c**. ITC measurements could not detect any interaction between LATS1 wild-type and non-phosphorylated MOB1, while phospho-MOB1 bound to LATS1 **d**. The NDR2(Y32V) mutant did not associate with non-phosphorylated MOB1, but bound to phospho-MOB1, although with a 20-fold decreased binding affinity compared to wild-type NDR2 (**e**). The mutation of Val647 to Tyr of LATS1 enabled LATS1 to bind to non-phosphorylated MOB1, and LATS1(V647Y) displayed a 5-fold increased binding affinity for phospho-MOB1 compared to wild-type LATS1 (**f**)

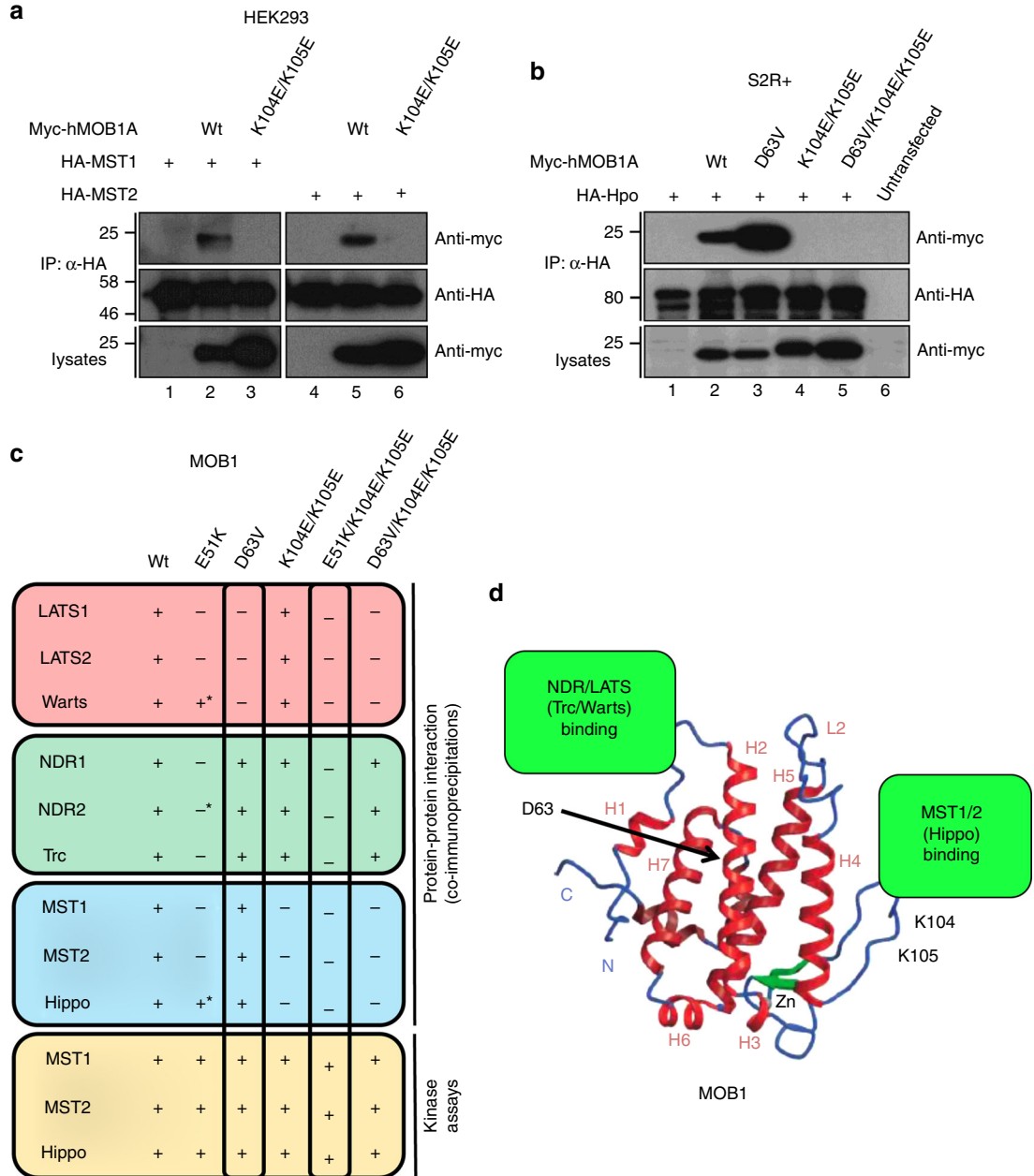

**Fig. 3** Definition of MOB1 variants with selective loss-of-interaction with Hippo core kinases. **a** Lysates of HEK293 cells expressing full-length HA-MST1 or HA-MST2 wild-type (*wt*) together with indicated full-length MOB1A versions were subjected to immunoprecipitation (*IP*) using anti-HA. Complexes were studied by immunoblotting using anti-myc (*top*) and anti-HA (*middle*). Input lysates were analyzed with anti-myc (*bottom*). The K104E/K105E mutations caused loss of binding to MST1/2. See Supplementary Fig. 19 for uncropped western blots. **b** Lysates of *Drosophila* S2R + cells expressing full-length HA-Hpo(wt) together with indicated full-length MOB1A versions were subjected to IP using anti-HA. Complexes were studied by immunoblotting using anti-myc (*top*) and anti-HA (*middle*). Input lysates were probed with anti-myc (*bottom*). Relative molecular weights are shown. See Supplementary Fig. 19 for uncropped western blots. **c** Schematic summary of the biochemical and molecular characterization of the indicated MOB1A variants presented in Supplementary Figs. 5–13. Noteworthy, NDR2 weakly interacted with MOB1(E51K), and Warts and Hpo bound normally to the E51K mutant as judged by co-immunoprecipitation experiments (marked by an *asterisk*), hence the E51K mutant was not studied further. **d** Model of human MOB1A(33–216) depicting the possibly opposing binding sites on MOB1 of NDR/LATS and MST1/2 kinases. Secondary structure elements of MOB1 are highlighted. The locations of Glu51, Asp63, and Lys104/Lys105 in MOB1 are indicated

cell survival based on the ability of single cells to grow into colonies[40]. Expression of all MOB1 variants resulted in decreased proliferation in 2D compared to controls (Fig. 4b). Likewise, except for MOB1(D63V/K104E/K105E), our MOB1 variants reduced colony formation (Fig. 4c, d). Transient expression of our MOB1 variants in HCT116 colon cancer cells also diminished proliferation and colony formation in 2D (Supplementary Fig. 14). These data show that MOB1(D63V) and MOB1(K104E/

K105E) suppress proliferation and colony formation similarly to MOB1(wt), suggesting that the interactions of MOB1 with LATS1/2 and MST1/2 are dispensable. Considering that MOB1 (D63V) and MOB1(K104E/K105E) still bind to NDR1/2 (Fig. 3c and Supplementary Fig. 6) and that MOB1(K104E/K105E) is phosphorylated on Thr12 and Thr35 in MCF-7 cells comparable to wild-type MOB1 (Supplementary Fig. 15), we are therefore tempted to conclude that MOB1 binding to NDR1/2 can be

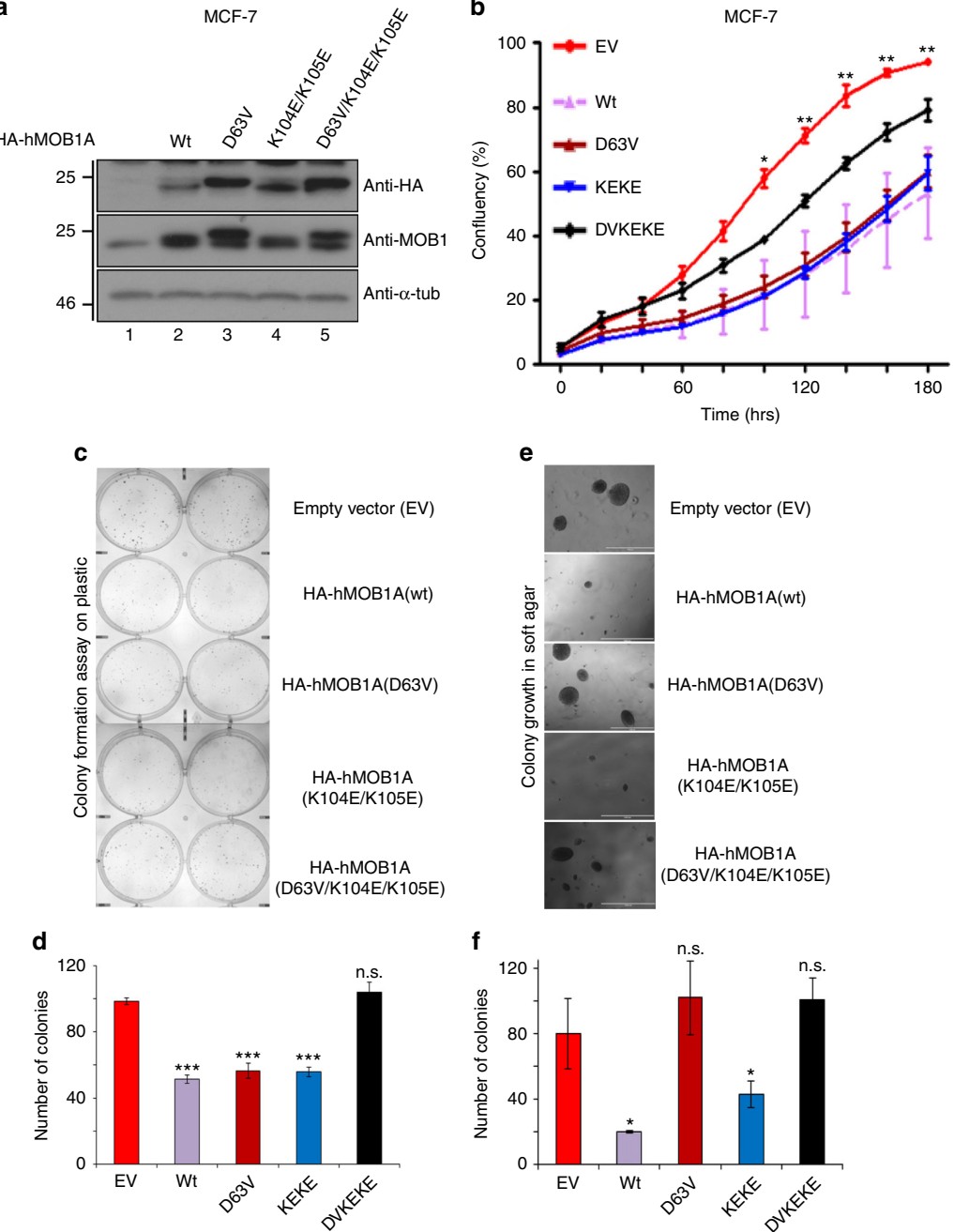

**Fig. 4** MOB1 binding to LATS1/2 is required to suppress anchorage-independent growth of human cancer cells, while MOB1 binding to MST1/2 is dispensable. **a** Immunoblotting with indicated antibodies of cell lysates derived from MCF-7 human breast cancer cells stably expressing the indicated HA-MOB1 variants or empty vector (*EV*) as negative control in lane 1. Relative molecular weights are indicated. See Supplementary Fig. 19 for uncropped western blots. **b** Proliferation rates of attached MCF-7 cells stably expressing indicated HA-MOB1A variants. The average of three experiments performed in triplicates with three independent cell pools is shown ($n = 3$). Statistically significant differences between EV and all MOB1 expressing cell pools are indicated (*$p < 0.05$; **$p < 0.01$). **c** Colony formation assays of the MCF-7 cells shown in **a**. Representative images are displayed. **d** Quantifications of colony formation assays shown in **c**. The average of three experiments performed in duplicates with three independent cell pools is shown ($n = 3$; ***$p < 0.001$; ns, not significant). *P*-values are: wt = 1.03E-05, DV = 5.42E-04, and KEKE = 1.95E-05 for comparison with EV cells. **e** Soft agar growth assays of the MCF-7 cells shown in **a**. Representative images are displayed. **f** Quantifications of the soft agar growth assays shown in **e**. The average of three experiments performed in duplicates with three independent cell pools is shown ($n = 3$; *$p < 0.05$; ns, not significant). *P*-values are: wt = 0.019 and KEKE = 0.04 for comparison with EV cells. *wt*, wild-type; *DV* D63V, *KEKE* K104E/K105E, *DVKEKE* D63V/K104E/K105E

sufficient to at least in part suppress cancer-related features in human cancer cells grown in 2D.

Next, we performed anchorage-independent growth assays (Fig. 4e, f), a more stringent method to determine malignant transformation in three-dimensional (3D) tissue culture[41]. In contrast to our 2D observations (Fig. 4b–d), MOB1(wt) or MOB1(K104E/K105E), but not MOB1(D63V) or MOB1(D63V/K104E/K105E), significantly suppressed anchorage-independent growth in 3D (Fig. 4e, f and Supplementary Fig. 14). This suggests that MOB1 interactions with MST1/2 are dispensable, while MOB1

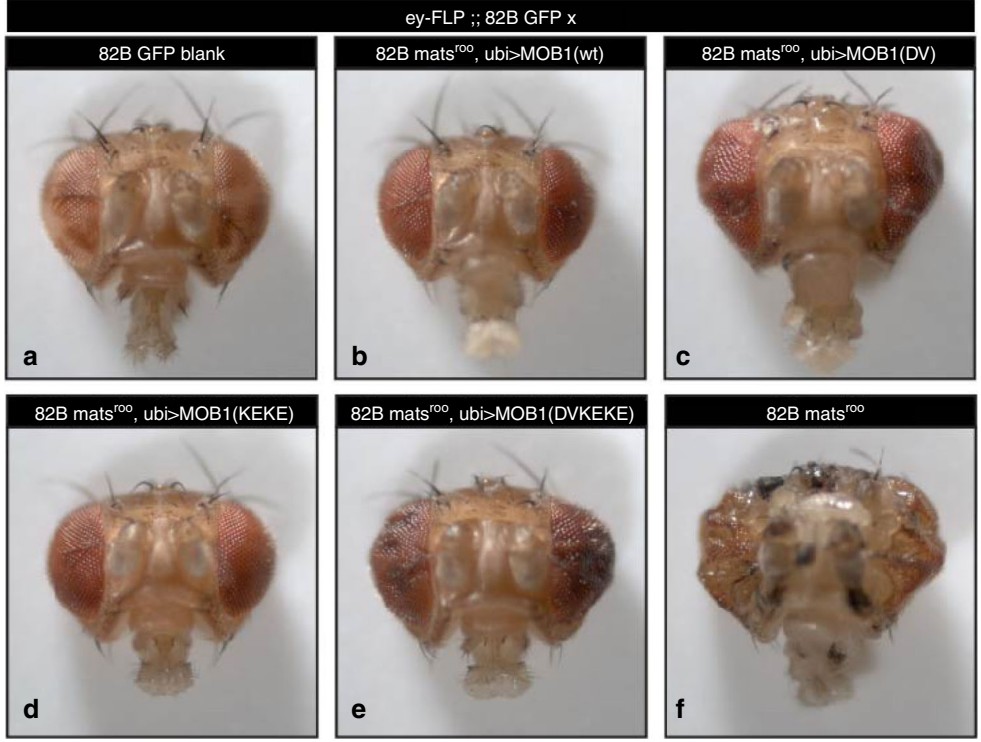

**Fig. 5** Expression of MOB1(wt) or MOB1(K104E/K105E), but not MOB1(D63V) or MOB1(D63V/K104E/K105E), fully suppresses the *mats* eye overgrowth phenotype. Shown are frontal views of eyes from female progeny of the indicated genotypes (*wt*, wild-type; *DV* D63V, *KEKE* K104E/K105E, *DVKEKE* D63V/K104E/K105E). A wild-type control eye is shown for comparison **a**. Expression of MOB1(wt) **b** or MOB1(K104E/K105E) **d** fully suppress the severe tissue overgrowth upon eyeless-FLP (*ey-FLP*) mediated formation of *mats* mutant clones in the eye (compare **b**, **d** with **a** and **f**). Expression of MOB1(D63V) **c** or MOB1(D63V/K104E/K105E) **e** does not fully suppress eye overgrowth upon *mats* loss-of-function (compare **c**, **e** with **a** and **f**)

binding to LATS1/2 is important and MOB1 binding to NDR1/2 alone is insufficient in this 3D setting. In general, our anchorage-independent growth data of human cancer cells (Fig. 4e, f) mirrored our fly genetics discoveries (see Fig. 5 and 6 and Supplementary Fig. 16 below), suggesting that tissue culture experiments performed under more physiological conditions can reflect in vivo tissue overgrowth experiments.

**MOB1/Hpo in fly development and tissue growth control.** To study our MOB1 variants in a complex multicellular organism, we generated and characterized transgenic flies that ubiquitously expressed our myc-tagged MOB1 versions in a *mats* mutant background (Figs. 5 and 6 and Supplementary Fig. 16). As human MOB1 expression can rescue *mats* mutants[10, 16], this allowed us to determine the functional significance of altering MOB1 binding in vivo. Using PhiC31-mediated recombination we integrated our MOB1 variants at the same chromosomal location (89E11 on chromosome 3) under control of the ubiquitous *ubiquitin-63E* promoter (*ubi > MOB1*, Supplementary Fig. 16a). Western blotting of whole flies confirmed similar expression of myc-tagged MOB1 variants (Supplementary Fig. 16b).

We then tested which MOB1 transgene can rescue the larval lethality of *mats* deficient flies[10]. As expected, wild-type MOB1 expression rescued the lethality of a null *mats* trans-heteroallelic combination (*mats^roo^/mats^e235^*, Supplementary Fig. 16c). Likewise, *mats* deficient animals expressing MOB1(K104E/K105E) were viable and fertile (Supplementary Fig. 16c), suggesting that stable MOB1/Hpo binding is dispensable for normal fly development. In contrast, neither MOB1(D63V) nor MOB1 (D63V/K104E/K105E) rescued *mats* mutants (Supplementary Fig. 16c), showing that MOB1/Wts complex formation is essential

for normal development, while MOB1 binding to Trc alone is insufficient to promote normal development.

To test the rescue effect of our MOB1 mutations on the *mats* tissue overgrowth phenotype, we generated *mats* mutant clones in the head using the eyFLP/FRT system[42]. Expression of wild-type MOB1 and MOB1(K104E/K105E) fully rescued the overgrown and misshapen head phenotype of eyFLP *mats* animals (compare Fig. 5b, d with Fig. 5a, f). In contrast, expression of MOB1(D63V) or MOB1(D63V/K104E/K105E) only partially suppressed the *mats* overgrowth phenotype (compare Fig. 5c, e with Fig. 5a, f). Thus, stable MOB1 binding to Hpo is dispensable for tissue growth control, while MOB1/Wts complex formation is necessary.

Finally, we tested the effect of our MOB1 transgenes on Yki transcriptional activity by examining the levels of Expanded (Ex), a well-characterized Yki transcriptional target[43]. We generated mutant clones for *mats* in wing imaginal disks (the larval precursors to the adult wing) using the FLP/FRT system under control of the heat shock promoter (Fig. 6). While *mats* clones displayed a robust increase in Ex expression (Fig. 6a–e) expression of either wild-type MOB1 or MOB1(K104E/K105E) restored Ex levels to control levels (Fig. 6f–j, p–t). In contrast, Ex levels (and therefore Yki activity) were still strongly upregulated when MOB1(D63V) or MOB1(D63V/K104E/K105E) were expressed in *mats* clones (Fig. 6k–o, u–y). Thus, in full agreement with the animal viability (Supplementary Fig. 16) and head overgrowth data (Fig. 5), the interaction of MOB1 with Wts is required to repress Yki activity, while the MOB1/Hpo interaction is dispensable and MOB1/Trc complex formation alone is insufficient for a complete rescue.

As previously observed[10], *mats* clones were usually small (Fig. 6a). *Mats* mutant clones were rarely recovered in the wing

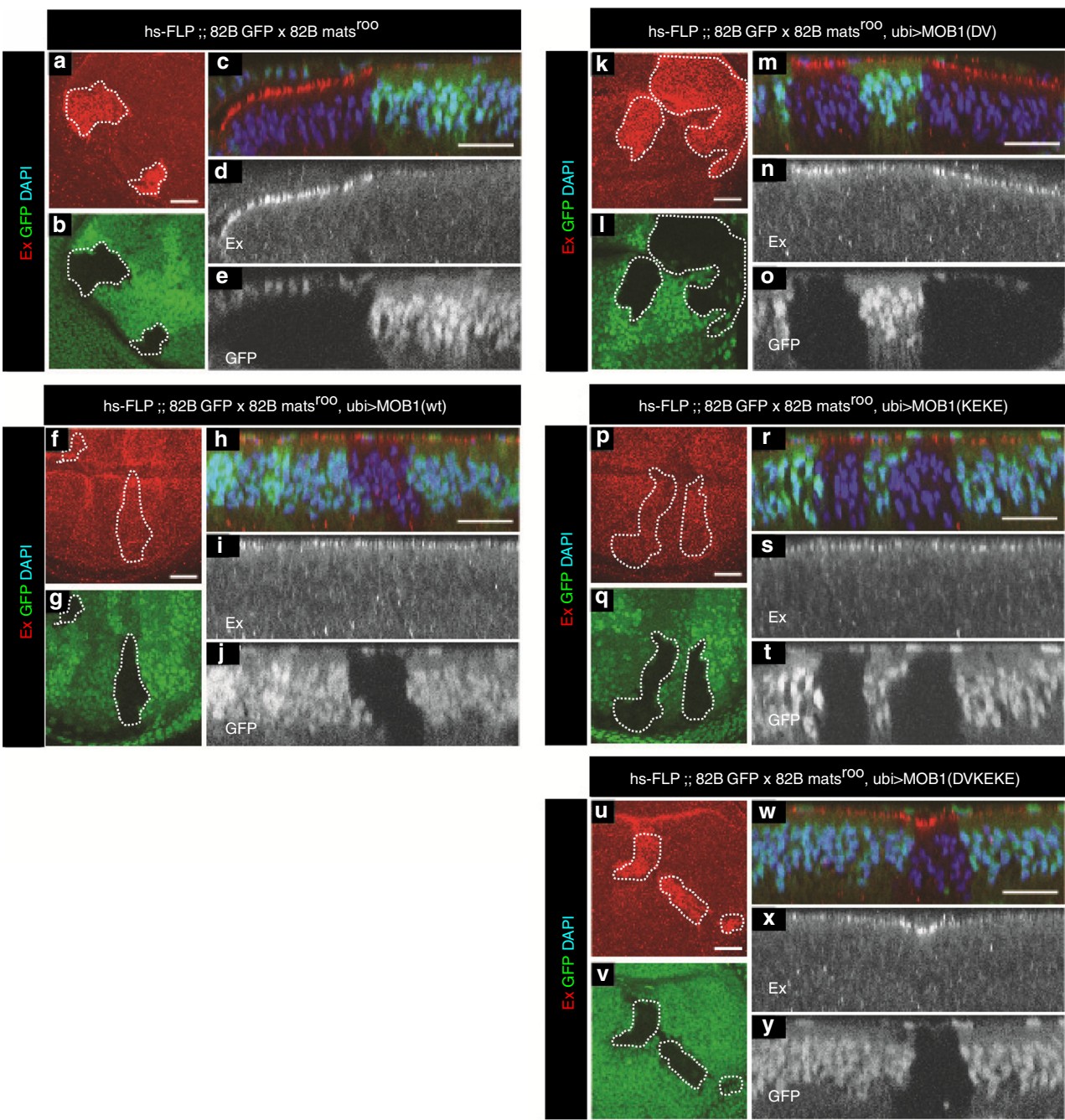

**Fig. 6** Human MOB1(wt) or MOB1(K104/K105E), but not MOB1(D63V) or MOB1(D63V/K104E/K105E), expression suppresses elevated Yki activity in *mats* null mutant wing imaginal disk clones. Analysis of Ex expression in wing imaginal disks of the indicated genotypes (*wt* wild-type, *DV* D63V, *KE* K104E/K105E, *DVKE* D63V/K104E/K105E). All panels show third instar wing imaginal disks containing *mats* mutant clones marked by the absence of GFP expression (*green* in **b**, **g**, **l**, **q**, **v**). GFP-negative *mats* null clones are indicated by *dashed lines*. Maximum intensity projections of XY confocal stacks are shown in *red* to visualize Expanded (*Ex*) labeling (*red* in **a**, **f**, **k**, **p**, **u**). X–Z confocal sections through the same wing imaginal disks were acquired **c**, **h**, **m**, **r**, **w**. Nuclei are stained with DAPI in blue. Ex (**d**, **i**, **n**, **s**, **x**) and GFP (**e**, **j**, **o**, **t**, **y**) protein levels are indicated. All scale bars are 20 μm. As expected, *mats* null clones displayed increased Yki activity as judged by elevated Ex levels **a–e**. Expression of MOB1(wt) **f–j** or MOB1(K104E/K105E) **p–t** suppresses Yki activity to normal levels in *mats* mutant clones (compare **i**, **s** with **d**), while expression of MOB1(D63V) **k–o** or MOB1(D63V/K104E/K105E) **u–y** does not (compare **n**, **x** with **d** and **i**)

pouch, likely because to their tendency to delaminate due to excessive overgrowth. Interestingly, although Ex levels were strongly upregulated in both MOB1(D63V) or MOB1(D63V/K104E/K105E) expressing clones, MOB1(D63V) expressing clones were larger and survived readily in the pouch (Fig. 6k), in contrast to MOB1(D63V/K104E/K105E) expressing clones, which were more similar to *mats* clones not expressing MOB1

(compare Fig. 6a, u). In this regard, we also noted that adult heads expressing MOB1(D63V/K104E/K105E) were more severely affected than adult heads expressing MOB1(D63V) in *mats* null tissue (compare the more rippled appearance of Fig. 5c, e). These findings collectively suggest that when MOB1 activity is weakened by loss of the MOB1/Wts interaction, MOB1 function is further compromised by loss of the Hpo/MOB1 interaction,

while disruption of the Hpo/MOB1 interaction alone does not affect MOB1 function.

## Discussion

Despite extensive progress in elucidating Hippo growth control signaling, studies comparing the biological significance of the regulatory interactions of MOB1 with Hippo core kinases have remained elusive. This point is crucial, since loss-of-function of MOB1 in flies and mice causes the most severe phenotypes of Hippo core cassette components[10, 23, 26, 27], indicating that MOB1 represents a multipurpose hub in Hippo core signaling. By combining genetics, structure, molecular, and cell biology our present study addresses this pressing issue. We uncovered key mechanisms promoting selective binding of MOB1 to Hippo core kinases in mammalian and fly cells. Precisely, we discovered that Asp63 of MOB1 is indispensable for interacting with LATS1/2 and Wts, while Lys104/Lys105 of MOB1 are essential for stable complex formation with MST1/2 and Hpo. Thus, MOB1 can differentiate between interactions in the Hippo core cassette, thereby most likely enabling MOB1 to regulate the specificity and amplitude of Hippo core kinase signaling.

While our biochemical and molecular data regarding Asp63 of MOB1 are further supported by a comparison of crystal structures of MOB1/NDR2 and MOB1/LATS2 complexes, we can currently only speculate on the structural level concerning Lys104/Lys105 of MOB1 and stable complex formation with MST1/2 (Hpo). Based on available structural data[28] one can draw the conclusion that Pro106 of MOB1 significantly contributes to the MST1 binding surface. Thus, we are tempted to speculate that modifications of Lys104/Lys105 of MOB1 impact the neighboring Pro106 and thereby impair stable MOB1/MST1 complex formation. Possibly, the positively charged Lys104/Lys105 residues of MOB1 further bond with the negatively charged phosphorylated Thr residues on MST1/2 (Hpo). In this regard, the available crystal structures[20, 28] of MOB1/MST1 and MOB1/MST2 complexes cover only a fraction of the possible interactions between MOB1 and different phospho-threonine residues on MST1/2. More specifically, these structures cover MOB1 bound to phosphorylated Thr353 and Thr367 of MST1 and phosphorylated Thr378 of MST2, while MOB1 binding to phosphorylated Thr329, Thr340, Thr380, and Thr387 of MST1 and phosphorylated Thr349, Thr356, and Thr364 of MST2 have been reported[20, 28]. The MOB1/MST1 and MOB1/MST2 crystal structures[20, 28] consistently highlighted Lys153 and R154 of MOB1 as central P0 phosphate coordinating residues of the phospho-threonine on MST1/2. Thus, we also tested the consequences of Lys153/R154 of MOB1, hoping to define an alternative MOB1 mutant that displays selective loss-of-interaction with human MST1/2 and fly Hpo. However, the testing of MOB1(K153A/R154A) and MOB1(K153E/R154E) revealed that Lys153/Arg154 of MOB1 are important for MST2 as well as LATS2 binding, supporting the previous notion that Lys153/Arg154 of MOB1 are likely to represent the core of a more general phospho-serine/threonine binding domain[37]. This interpretation is further supported by the finding that Lys153/Arg154 of MOB1 are part of the Praja2 binding region[38].

Our data indicate that MOB1 binding to the NTRs of NDR1/2 vs. LATS1/2 also differs quantitatively. By determining interaction affinities, we observed an interaction of non-phosphorylated MOB1 with NDR2, while binding to LATS1 was not detectable. Phospho-MOB1 bound to LATS1, but interacted with NDR2 at a much higher affinity. These findings suggest that MOB1 generally displays a significantly higher affinity for NDR2, hence MOB1-mediated Hippo signaling may preferentially signal through the NDR kinase branch, although our biological data indicate that MOB1/NDR (Trc) complex formation alone is insufficient to support development and normal tissue growth control. In this regard, His646 and Val647 of LATS1 (see this study), and possibly selective inhibitory MOB2 binding to the NTR of NDR1/2[44], are promising candidates for the fine-tuning of MOB1-mediated signaling. Consequently, future studies are warranted to address these possibilities. In particular, crystal structures of full-length NDR kinases bound to full-length MOB1 in its non-phosphorylated vs. phosphorylated state will help to further our understanding with regard to the recently proposed auto-inhibition model for MOB1 binding[19, 20]. Furthermore, the in vivo importance of MST1/2 (Hpo) mediated phosphorylation of MOB1 (Mats) needs to be deciphered. In this regard, our conclusions regarding MOB1 phosphorylation are currently based on in vitro experiments, which must be cautiously interpreted regarding in vivo implications.

By discovering MOB1 variants displaying selective loss-of-interactions and decrypting the biological significance of regulatory interactions of MOB1 in Hippo core signaling, we believe that our study helps to settle the recent controversy[3, 19, 20, 30, 31] regarding the importance of MOB1 binding to MST1/2 (Hpo). Ni et al. previously concluded that stable MST2 binding to MOB1 functions as an important step in activating the MST1/2-LATS1/2 kinase cascade[20]. However, no biological functions were addressed[20]. In this regard, using a chimeric sensor to measure Wts conformation in fly tissues, Vrabioiu and Struhl[31] found that MOB1 can act as a Hpo-independent activator of Wts, hence contrasting the model proposed by Ni et al.[20]. Manning and Harvey[30] proposed a unifying model, wherein MOB1 acts before and after Hpo-mediated phosphorylation of Wts and MOB1. However, in support of Luo and colleagues[20], the Sicheri and Gingras laboratories recently showed that ternary MST1/2-MOB1-LATS1/2 (NDR1/2) complex formation is important, at least when tested in vitro[28, 29]. Conversely, our study rather supports the model drawn by Vrabioiu and Struhl[31], namely that a ternary complex is not required for MOB1/LATS1 activation in vivo. Therefore, it is currently difficult to reconcile all published data into one general model. Nonetheless, we are proposing an updated four-step model (Supplementary Fig. 17) attempting to consolidate these models[20, 30, 31] with our discoveries reported here and other important biochemical data[19, 21, 22, 28, 29, 35, 36]. First, activated Hpo (MST1/2) phosphorylates MOB1 to release MOB1 from an auto-inhibitory conformation[19, 20, 22]. Significantly, this first step does not seem to require formation of a stable Hpo/MOB1 complex (see this study), suggesting that MST1/2 phosphorylation of MOB1 only requires a brief transient kinase-substrate interaction. However, we currently do not understand how MOB1 (Mats) phosphorylation actually fits into the regulation of Hippo core kinase signaling in vivo. Second, MOB1 binds to Wts (LATS1/2) to "open up" Wts[31]. In this second step the formation of a stable MOB1/Wts complex is essential (see this study and refs. [21, 22, 35, 36]). Third, "open" LATS1/2 is phosphorylated by MST1/2 in the C-terminal hydrophobic motif (HM)[20, 21]. This third step can occur without formation of a stable ternary MST1/2-MOB1-LATS1/2 complex, as proposed by the MST1/2-binding deficient K104E/K105E mutant characterized in this study. Fourth, HM-phosphorylated LATS1/2 autophosphorylates on the activation loop[20, 21], a step that can occur independent of MOB1 binding to LATS1/2[21]. Intriguingly, the MST1/2-MOB1-NDR1/2 signaling model is very similar, but different[17, 45], since MOB1 binding to NDR1/2 can occur independently of MOB1 phosphorylation (see this study). Even more importantly, it is crucial to note that our proposed model (Supplementary Fig. 17) is mainly supported by in vitro experiments.

Our study together with the report by Vrabioiu and Struhl[31] would argue that the MOB1/Hippo interaction is dispensable for tissue growth control by Hippo signaling. But we strongly caution from drawing a broad and general conclusion from these studies, since we cannot rule out the possibility that the MOB1/Hippo interaction is only dispensable in selective aspects of tissue growth control. Thus, our model (Supplementary Fig. 17) may exemplify a mechanism for a switch-like activation of NDR/LATS kinases in specific biological contexts (for example, to ensure abrupt termination of growth responses), consequently being limited to specific biological settings. As a result, much more future research is warranted to further dissect these subtle, but important, differences in the activation mechanisms of LATS1/2 vs. NDR1/2 kinases in specific physiological contexts.

Surprisingly, our study further revealed that stable MOB1/Hpo interaction is dispensable for development, tissue growth control and suppression of Yki activity. We also discovered that the MOB1/Trc interaction alone is insufficient to normally support these processes, although it can be sufficient to decrease proliferation of human cancer cells. The MOB1/Wts interaction is essential for development, tissue growth control, and Yki regulation in *Drosophila*, but it can be dispensable for some tumor suppressive properties of MOB1 in human cells. Therefore, our study significantly advances our understanding of the biological importance of the regulatory interactions of MOB1 with Hippo core kinases, in addition to providing structural and molecular insights into the differential binding of MOB1 to Hippo core kinases.

In the course of our in vivo studies we noted further interesting aspects. While disruption of the stable MOB1/Hpo interaction alone did not have a detectable effect on MOB1 function (see K104E/K105E mutant), loss of the MOB1/Hpo interaction could affect MOB1 function in the context of disrupted MOB1/Wts interaction (see D63V/K104E/K105E mutant). This is illustrated by the observations that, in the eye overgrowth assay, D63V/K104E/K105E mutant tissues were noticeably more overgrown than in K104E/K105E mutants. In the wing clone experiments, only D63V/K104E/K105E mutants displayed a high frequency of clone delamination and loss in the wing pouch, indicative of a strong overgrowth phenotype and similar to the full *mats* mutant phenotype. Thus, in the context of a weakened MOB1 function (i.e. through disrupted MOB1/Wts binding), loss of the MOB1/Hpo interaction can further reduce the in vivo function of MOB1.

Another interesting aspect based on our in vivo work is that, although D63V/K104E/K105E rescued flies showed a strong eye overgrowth phenotype and Ex upregulation, these phenotypes were still markedly weaker than the full *mats* loss-of-function phenotypes, suggesting that D63V/K104E/K105E can at least partially rescue the *mats* mutant phenotype. This could be due to several reasons. First, as the MOB1/Trc interaction is not disrupted in the D63V/K104E/K105E mutant, this mutant may partially rescue the *mats* mutant phenotype through MOB1-mediated Trc regulation. Indeed, Trc has been proposed to function partially redundantly with Wts, at least in some contexts[46]. Second, although our data indicate that the D63V/K104E/K105E mutant is severely impaired in its ability to bind to Wts, it is nevertheless possible that some low-affinity Wts binding activity might remain, which possibly is sufficient to partly rescue the *mats* mutant phenotype. Third, we cannot exclude the possibility that other factors, besides the Hippo core kinases, may facilitate MOB1-mediated Hippo signaling. In this regard, other binding partners of MOB1 are worth considering, although the known MOB1 binders Praja2[38] and Dock8[47] are not conserved in flies. Therefore, future experiments are warranted to address this issue, taking into account recent interactome screens[29, 48, 49].

Taken together, our study establishes a major foundation of MOB1-mediated Hippo core signaling. The Hippo pathway is essential for tissue growth control and homeostasis[1–3], and its dysregulation has been linked to various human cancers[4]. Therefore, our study through providing notable insights into how MOB1 differentiates between Hippo core kinase defines a framework for how these different interactions may function in different cellular contexts in health and disease. We discovered that selected regulatory interactions of MOB1 are essential for development, tissue growth control, and Yki regulation, hence establishing MOB1 as the central hub of Hippo core signaling, besides providing structural and molecular insights into the Hippo core cassette, and establishing key research tools for future studies of the regulatory interactions of MOB1 in diverse disease-relevant settings.

## Methods

**Protein purification for structural and biochemical analyses.** cDNAs encoding human MOB1A (residues 33–216), NDR1 (residues 24–87), NDR2 (residues 25–88), LATS1 (residues 618–697 or 640–703), and MST2 (residues 2–392 or 2–308) were subcloned into a modified pET28a vector (Novagen). NDR1 (24–87, Y31V), NDR2 (25–88, Y32V), and LATS1 (640–703, V647Y) were generated by standard PCR mutagenesis. Human LATS2 (residues 603–666) and LATS2 (603–666, V610Y) were codon-optimized, synthesized, and subcloned into a modified pET28a vector (Novagen) by Generay Biotechnology. These pET28a-based constructs were used to express N-terminally 6× His tagged proteins in the *E. coli* strain BL21(DE3). BL21(DE3) were from New England BioLabs. Bacteria carrying pET28a plasmids were grown at 37 °C until they reached an optical density of 0.6–1.0 at 600 nm. Protein expression was induced by addition of 0.2 mM IPTG (Isopropyl β-D-1-thiogalactopyranoside) and cultures were incubated at 16 °C for additional 12–16 h. After cell lysis with a cell homogenizer (JNBIO) in lysis buffer (25 mM Tris pH 8.0, 300 mM NaCl, and 20 mM imidazole), followed by centrifugation, supernatants were subjected to $Ni^{2+}$-NTA affinity chromatography (Qiagen) as described by the manufacturer. To form crystals of the MOB1A/NDR2 complex, NDR2 (25–88) and MOB1A (33–216) proteins were incubated at an 1:1 molar ratio at 4 °C in assembly buffer (20 mM Tris pH 8.0, 200 mM NaCl, and 5 mM DTT (Dithiothreitol)), prior to loading onto a HiLoad 16/60 Superdex 200 gel filtration column (GE Healthcare). Peak fractions were combined and concentrated to 10–15 mg ml$^{-1}$ for crystallization experiments. Purified MOB1A (33–216) was phosphorylated as follows: in a total volume of 10 ml MOB1A (1 mg ml$^{-1}$) was incubated with MST2 (2–308, 0.002 mg ml$^{-1}$) at 30 °C for 90 min in phosphorylation buffer (40 mM Tris pH 7.5, 10 mM MgCl$_2$, 5 mM DTT, and 0.2 mM ATP). Completion of the phosphorylation reaction was verified by the shift of the MOB1A band in non-denaturing gel electrophoresis conditions.

**Crystallization and data collection.** Crystals of MOB1A (33–216) bound to NDR2 (25–88) were grown at 14 °C in crystallization buffer (17% PEG 3350, 0.1 M HEPES pH 7.5, and 150 mM magnesium chloride) by the hanging-drop vapor-diffusion method. Crystals were transferred to the crystallization buffer supplemented with 25% glycerol before freezing. Using an ADSC Quantum 315r CCD area detector an X-ray diffraction data-set at was collected at the beamline BL17U1 at Shanghai Synchrotron Radiation Facility (Shanghai, China). Diffraction data 2.10 Å were processed with the HKL2000 software[50].

**Structure refinement and molecular graphics.** The crystal belonged to the space group $P2_12_12_1$, and contained two MOB1A (33–216)-NDR2 (25–88) complexes in each asymmetric unit. MOB1 (33–216) molecules in the complex were located by the molecular replacement method with the CCP4 program Phaser[51, 52] using the published[32] structure of MOB1A (33–216) alone (PDB code: 1PI1) as the searching model. The model of the NDR2 (25–88) fragment was built manually with Coot[53]. After refinement by the CCP4 program REFMAC[54], the $R_{work}$ and $R_{free}$ factors were 15.3 and 24.4%, respectively. The quality of the final structure was verified by the CCP4 program PROCHECK[51], revealing good stereochemistry according to the Ramachandran plot (99.0, 0.6, and 0.4% for favored, allowed, and disallowed regions, respectively). The final models include residues 33–212 and 33–216 of MOB1A for chains A and B, respectively, and residues 25–85 and 25–87 of NDR2 for chains C and D, respectively. All figures displaying protein structures were generated with PyMOL (http://www.pymol.org). Sequence conservation of MOB1 mapped onto the surface of its crystal structure was generated by the ConSurf server[55].

**Isothermal titration calorimetry assays.** ITC experiments were performed at 25 °C using an ITC200 system (MicroCal) and the following buffer system: 25 mM HEPES pH 7.5, 200 mM NaCl, and 1 mM EDTA. Proteins were centrifuged and degassed before experiments. Typically, ~240 μM MOB1A was injected 20 times in 2 μl aliquots into a 200 μl sample containing ~24 μM of NDR1, NDR2, LATS1, or

LATS2 proteins in the sample cell. By varying the stoichiometry ($n$), the enthalpy for the reaction ($\Delta H$), and the association constant ($K_a$), data were fitted with the non-linear least-square method using a single-site binding model with Origin for ITC version 7.0 (MicroCal).

**Gel filtration chromatography.** Full-length MOB1A wild-type, D63V or K104E/K104E cDNAs were subcloned into the pET28a vector, expressed in bacteria and subsequently purified as described above for MOB1A (33–216). Purified full-length MOB1 proteins were incubated with NDR2 (25–88), LATS1 (618–697), or MST2 (2–392) protein at 4 °C for 20 min in assembly buffer, before the protein mixtures were loaded onto a HiLoad 16/60 Superdex 200 gel filtration chromatography column (GE Healthcare) and 1.4 ml fractions were collected. Samples from selected Superdex 200 fractions were analyzed by sodium dodecyl sulfate-polyacrylamide gel electrophoresis (SDS-PAGE) and Coomassie Blue staining. For the complex formation assay between MOB1A and LATS1, MOB1A protein was in vitro phosphorylated by MST2 (2-308) as described above prior to incubation with LATS1 (618–697).

**Antibodies for immunoblotting and immunoprecipitations.** For immunoblotting, samples were resolved by 8 or 12% SDS-PAGE, followed by transfer onto Immobilon-P membranes (Millipore). Membranes were blocked for at least one hour with TBST (50 mM Tris pH 7.5, 150 mM NaCl, 0.5% Tween 20) containing 5% skim milk powder and then probed with primary antibody. Bound antibodies were detected by horseradish peroxidase-linked secondary antibodies, followed by enhanced chemiluminescence. For co-immunoprecipitations, cells were collected by centrifugation at $1000 \times g$ for 3 min, followed by lysis in corresponding buffers (see below). After 30 min, cell lysates were centrifuged for 10 min at $16,000 \times g$ at 4 °C before preclearing with protein A-Sepharose, followed by immunoprecipitation with 12CA5 antibody pre-bound to protein A-Sepharose. Beads were washed at least three times with the corresponding lysis buffer before samples were analyzed by SDS-PAGE and immunoblotting. The characterization of LATS1/2, NDR1/2, MST1/2, Warts, Trc, and Hpo binding to MOB1A variants was carried out in low-stringency buffer (30 mM HEPES pH 7.4, 20 mM beta-glycerophosphate, 20 mM KCl, 1% TX-100, 1 mM EGTA, 2 mM NaF, 1 mM $Na_3VO_4$, 1 mM benzamidine, 4 µM leupeptin, 0.5 mM PMSF, 1 µM microcystin, and 1 mM DTT). Alternatively (Supplementary Fig. 8), co-immunoprecipitation experiments were performed in RIPA (9806, Cell Signaling; 20 mM Tris pH 7.5, 150 mM NaCl, 1 mM EDTA, 1 mM EGTA, 1% NP-40, 1% (w/v) sodium deoxycholate, 2.5 mM sodium pyrophosphate, 1 mM β-glycerophosphate, 1 mM $Na_3VO_4$, 1 µg ml$^{-1}$ leupeptin, 1 mM benzamidine, 0.5 mM PMSF, 1 µM microcystin, and 1 mM DTT), PBS-E (phosphate-buffered saline (10 mM $Na_2HPO_4$, 2 mM $KH_2PO_4$, 137 mM NaCl, 2.7 mM KCl at pH 7.4) supplemented with 1 mM EDTA, 1% (v/v) Triton X-100, 50 mM NaF, 1 mM $Na_3VO_4$, 1 mM benzamidine, 4 µM leupeptin, 0.5 mM PMSF, 1 µM microcystin, and 1 mM DTT) or standard lysis buffer (20 mM Tris pH 8.0, 150 mM NaCl, 10% glycerol, 1% NP-40, 5 mM EDTA, 0.5 mM EGTA, 20 mM β-glycerophosphate, 50 mM NaF, 1 mM $Na_3VO_4$, 1 mM benzamidine, 4 µM leupeptin, 0.5 mM PMSF, 1 µM microcystin, and 1 mM DTT). Anti-GAPDH (GTX100118 from GeneTex; used at 1:1,000 on fly extracts) confirmed equal loading. Anti-HA 12CA5 (used at 1:500), anti-myc 9E10 (used at 1:50), anti-α-tubulin YL1/2 (used at 1:100), and anti-MOB1 (used at 1:250) antibodies have been described[39, 45, 56, 57]. Additional anti-HA antibodies were from Cell Signaling (C29F4; used at 1:1000) and Roche (3F10; used at 1:1000). Anti-myc (71D10; used at 1:1000), anti-MST1 (3682; used at 1:2500), anti-MST2 (3952; used at 1:2500), anti-T12-P (8843; used at 1:1000), and anti-T35-P (8699; used at 1:1000) were from Cell Signaling. Anti-actin (sc-1616; used at 1:1000) were from Santa Cruz Biotechnology. Anti-GAPDH (Mab374; used at 1:1000), anti-Mal (Maltose binding protein; E8032; used at 1:5000), and anti-GST (ab6613; used at 1:5000) were from Millipore, New England BioLabs, and Abcam, respectively. Secondary antibodies were purchased from GE Healthcare (NA931, NA934, and NA935; used at 1:5000 and 1:10,000, respectively) and Santa Cruz Biotechnology (sc-2020; used at 1:5000). Primary and secondary antibodies for immunofluorescence studies (all used at 1:500) were rabbit anti-Expanded (kindly provided by A. Laughon, University of Wisconsin–Madison, USA), mouse anti-β-gal (Promega, Z3781), Rhodamine Red X-conjugated anti-rabbit and FITC-conjugated anti-mouse antibodies (both from Jackson ImmunoResearch).

**Kinase assays.** Human full-length MOB1A cDNA variants were inserted into the pMal-c2 vector (New England BioLabs) using *Bam*HI and *Xho*I/*Sal*I sites to generate pMal-MOB1A plasmids, which can express MOB1A N-terminally tagged by the maltose binding protein. Recombinant Mal-MOB1A proteins were expressed in *E. coli* BL21(DE3) at 30 °C and purified using amylose resin (E8021, New England BioLabs). Briefly, single-bacteria colonies containing pMal-MOB1A were grown overnight at 37 °C, followed by 1:10 dilution before IPTG was added at 100 µM and bacteria were incubated for another 3 h at 30 °C. Subsequently, cultures were centrifuged at $1400 \times g$ for 10 min, followed by addition of Mal lysis buffer (20 mM Tris pH 7.5, 1 mM EDTA, 200 mM NaCl) and sonication at 70% amplitude (model:VCx 130, Sonics Vibra Cell). Lysates were centrifuged and the supernatant was clarified by filtration (40 µm cell strainer) before addition of amylose-resin and overnight incubation at 4 °C. The following day, the protein-amylose resin mix was washed excessively with Mal lysis buffer, followed by elution of Mal-MOB1A proteins in 20 mM Tris pH 7.5 buffer containing 10 mM maltose. Recombinant Mal-MOB1A proteins were dialyzed in 20 mM Tris pH 7.5 containing 10% (v/v) glycerol, followed by quantification using SDS-PAGE and Coomassie staining. Recombinant full-length wild-type GST-MST1 (M9697) and GST-MST2 (S6573) were from Sigma. To produce immunopurified full-length wild-type HA-tagged Hpo kinase, *Drosophila* S2R + cells were transfected with pAW_HA-Hpo and processed for immunoprecipitation (IP) using anti-HA antibody and standard IP conditions as defined above. Immunopurified Hpo proteins were washed twice with MST1/2 (Hpo) kinase buffer (5 mM Tris pH 7.5, 2.5 mM beta-glycerophosphate, 1 mM EGTA, 1 mM $Na_3VO_4$, 4 mM $MgCl_2$, 0.1 mM DTT), before kinase reactions were performed as follows: per reaction 200 ng of Mal-MOB1A proteins were incubated at 30 °C for 30 min in 20 µl of kinase reaction buffer (5 mM Tris pH 7.5, 100 µM ATP, 2.5 mM beta-glycerophosphate, 1 mM EGTA, 1 mM $Na_3VO_4$, 4 mM $MgCl_2$, 0.1 mM DTT) in the absence or presence of GST-MST1/2 (50 ng per reaction) or immunopurified Hpo. The reactions were stopped by the addition of Laemmli buffer, before proteins were separated by SDS-PAGE, followed by immunoblotting as outlined above.

**Construction of plasmids.** Human MOB1A/B, NDR1/2, LATS1/2, and MST1/2 cDNAs cloned in pcDNA3-based vectors were described previously[21, 35, 39, 44, 45, 56]. pcDNA3_myc-hMOB1A(wt) served as template for the generation of the following MOB1A mutants by PCR-based mutagenesis: E49R, E51K, E55K, D63V, K104E, K105E, K135E, H161Q, H164Q, K104E/K105E, D127K/D128K, K153A/R154A, K153E/R154E, E51K/K104E/K105E, and D63V/K104E/K105E. To subclone N-terminally HA-tagged MOB1 cDNAs into the pLEX vector, the tagged cDNAs were first inserted using *Kpn*I and *Xho*I into the pENTR-3C plasmid (Invitrogen) and then recombined into the pLEX destination plasmid using Gateway technology (Invitrogen). To subclone HA-tagged MOB1 cDNAs into the pCMV-R-neo plasmid[58], the tagged cDNAs were inserted using *Pme*I and *Xho*I. Hpo and Warts cDNAs have been described[59]. The pAW vector, Mats (LD47533) and Trc (LD37189) cDNAs were from the Drosophila Genomics Resource Center (Indiana University, USA). To subclone N-terminally tagged cDNAs into the pAW or pKC26w-pUbiq fly expression vectors, the tagged cDNAs were first inserted into the pENTR-3C plasmid (Invitrogen) and then recombined into these two destination plasmids using Gateway technology (Invitrogen). Myc-hMOB1A variants, myc-Mats, HA-Trc, and HA-Hpo were inserted into pENTR-3C using *Kpn*I and *Xho*I. HA-Warts was inserted into pENTR-3C using *Kpn*I and *Not*I. All constructs were confirmed by sequence analysis of the entire cDNAs at every cloning step.

**Cell culture and transient transfections.** HEK293, HEK293T, MCF-7, and HCT116 cells were originally obtained from ATCC and grown at 37 °C in 5% $CO_2$ humified chambers in DMEM (D6429, Sigma) supplemented with 10% fetal bovine serum (FBS; F7524, Sigma) and penicillin/streptomycin. Cells were authenticated through STR (Short Tandem Repeat) profiling and mycoplasma tested by Microsynth (Switzerland). Exponentially growing HEK293 cells were plated at a consistent confluence and transfected with plasmids using Fugene 6 (E2692, Promega) according to the manufacturer's instructions as described[45, 60]. $1.2 \times 10$ E6 of HCT116 cells were transiently transfected with 1.0 µg pcDNA3-based plasmids using the nucleofector kit V (VCA-1003, Lonza) as defined by the manufacturer. *Drosophila* S2R + cells were maintained at 24 °C in Schneider's *Drosophila* Medium (217200024, Invitrogen) supplemented with heat inactivated FBS (10082147, Invitrogen) and penicillin/streptomycin. S2R + cells were transiently transfected with pAW-based plasmids using Effectene (301425, Qiagen) according to the manufacturer's instructions.

**Generation of stable cell line pools.** To generate stable lentiviral cell pools using pLEX_HA-MOB1 plasmids, $8 \times 10$ E6 of the HEK293T packaging cells were transfected with 2.4 µg of pMD.G, 0.8 µg of p8.91, and 4.8 µg of pLEX plasmids using Lipofectamine 2000 (11668, Invitrogen) as recommended by the manufacturer. Tissue culture supernatants were harvested 24 h later, passed through a 0.45-µm filter and added to the target cell lines in the presence of 1 µg ml$^{-1}$ polybrene. Infected cells were selected by growth in the presence of 2.5 µg ml$^{-1}$ puromycin. Stable pools (uncloned mass culture) of cells were maintained in DMEM supplemented with 10% FBS and 0.6 µg ml$^{-1}$ puromycin. Retroviral pools (uncloned mass culture) were generated using pCMV-R-retro based plasmids and maintained in the presence of G418 as described elsewhere[58, 61].

**Proliferation and colony formation assays.** HCT116 transiently transfected with pcDNA3-based plasmids were seeded 24 h post-transfection, or stable MCF-7 cell line pools were analyzed. Media were replenished every 72 h during the duration of experiments. For cell proliferation analysis, cells were seeded at defined densities ($5 \times 10$ E4 cells per well) in triplicates in 12-well plates, followed by non-invasive IncuCyte live cell imaging (Essen BioScience) to measure the kinetics of cell growth/proliferation based on area (confluence) metrics. Phase-contrast images were continuously collected on IncuCyte ZOOM (Essen BioScience) for at least one week. A specific processing definition (Phase-contrast processing module) was applied to count objects (cells) for the duration of the assay. The Phase object area was expressed as relative cell confluency for each well at each time point and

subsequently exported into GraphPad Prism Software for final analysis. To evaluate colony formation, 1,000 cells were seeded per well (6-well format). After 8–12 days, colonies were fixed with methanol/acidic acid (3:1) for 5 min at room temperature, stained with 0.5% (w/v) crystal violet for 15 min at room temperature, washed with water and finally scanned using the G:BOX HR gel documentation system (Syngene). Colonies composed of at least 50 cells were score as positive.

**Soft agar assays for anchorage-independent growth.** After trypsinization, cells were passed 4-5 times through a 21 G syringe, before $1 \times 10$ E4 were resuspended in complete medium (DMEM containing 10% FBS and appropriate antibiotics) with 0.6% agarose (16520050, Thermo Fisher Scientific), and subsequently cultured in wells (6-well format) underlaid by a layer of 1% agarose in complete medium and overlaid with complete medium without agarose. Top layer media were replenished every 72 h. After three weeks, colonies were stained with 2.5 mg ml$^{-1}$ thiazolyl blue tetrazolium blue (MTT, Sigma), scanned, and quantified. Cell clusters of at least 50 cells were scored as colonies.

**Drosophila melanogaster genetics.** All flies were maintained at 25 °C unless otherwise stated. The *mats$^{roo}$* and *mats$^{e235}$* fly strains, carrying *mats* null alleles, have been reported[10]. The *mats* gene is located on chromosome 3 at 94A12 (FlyBase ID: FBgn0038965). The PhiC31 integrase-mediated site-specific recombination method was used to express human MOB1 variants from one identical chromosomal location (PBac{y[ + ]-attP-9A}VK00027 integrated at 89E11 on chromosome 3) (see also Supplementary Fig. 16a). Briefly, N-terminally myc-tagged MOB1 cDNA variants were cloned into the pKC26w-pUbiq vector that allows expression of the cloned fragments under the *ubiquitin-63E* promoter[62]. pKC26w-pUbiq_myc-MOB1 plasmids were injected into y[1] w[1118]; PBac{y[ + ]-attP-9A}VK00027 flies (FlyBase ID: FBst0009744) and stable transformants were identified by the presence of the *mini-white$^+$* marker. Injections of embryos with pKC26w_pUbiq _myc-hMOB1A plasmids and the initial generation of stable transformants were performed by BestGene Inc. (Chino Hills, USA). The *ubi > MOB1* transgenes were recombined with the *mats$^{roo}$* allele on an *FRT82B*-carrying chromosome by meiotic recombination followed by selection in medium containing 100 μl of 25 mg ml$^{-1}$ neomycin. Upon appearance of eggs, vials were heat-shocked at 37 °C for 1 h 1–2 times every day until pupae appeared. After balancing, the *mats$^{roo}$* allele and FRT82B sites were confirmed by PCR genotyping (see below). To generate adult eyes mosaic for homozygous mutant and heterozygous wild-type tissue, these recombinant lines were crossed to *eyFLP;; FRT82B GFP/TM6B* (*eyFLP* expresses FLP only in the eye and head capsule[42]). To generate mutant clones in wing imaginal disks, we crossed our lines to *hsFLP; FRT82B GFP*. Larvae were heat-shocked at 37 °C for 1 h 3 days after egg laying, followed by dissection at wandering L3 stage (6 or 7 days after egg laying).

**Genotypes of transgenic flies used in this study.** *Figure 5*:

a. *yw eyFLP/+ ;; FRT82B GFP/FRT82B blank*
b. *yw eyFLP/+ ;; FRT82B GFP/FRT82B mats$^{roo}$, ubi > MOB1-wt*
c. *yw eyFLP/+ ;; FRT82B GFP/FRT82B mats$^{roo}$, ubi > MOB1-DV*
d. *yw eyFLP/+ ;; FRT82B GFP/FRT82B mats$^{roo}$, ubi > MOB1-KEKE*
e. *yw eyFLP/+ ;; FRT82B GFP/FRT82B mats$^{roo}$, ubi > MOB1-DVKEKE*
f. *yw eyFLP/+ ;; FRT82B GFP/FRT82B mats$^{roo}$*

*Figure 6*:

a-e. *yw hsFLP/+ ;; FRT82B GFP/FRT82B mats$^{roo}$*
f-j. *yw hsFLP/+ ;; FRT82B GFP/FRT82B mats$^{roo}$, ubi > MOB1-wt*
k-o. *yw hsFLP/+ ;; FRT82B GFP/FRT82B mats$^{roo}$, ubi > MOB1-DV*
p-t. *yw hsFLP/+ ;; FRT82B GFP/FRT82B mats$^{roo}$, ubi > MOB1-KEKE*
u-y. *yw hsFLP/+ ;; FRT82B GFP/FRT82B mats$^{roo}$, ubi > MOB1-DVKEKE*

**Immunofluorescensce microscopy of *Drosophila* imaginal disks.** All procedures were carried out at room temperature, protected from light, unless otherwise stated. Wandering L3 larvae were dissected in ice-cold phosphate-buffered saline (PBS). Briefly, larvae were torn in half, the anterior half inverted, and undesired tissues were removed (central nervous system, fat body, and salivary glands), leaving only the imaginal disks attached to the inverted carcass. Carcasses were then fixed in 4% paraformaldehyde in PBS at room temperature for 30 min, washed several times in PBS containing 0.1% Triton-X100 (PBST), followed by permeabilization for 25 min in PBS containing 0.3% Triton-X100 and 1 h pre-blocking with 10% normal goat serum (NGS) in PBST. Subsequently, samples were incubated with primary antibody at 4 °C overnight. Then samples were washed with PBST, pre-blocked once more with 10% NGS in PBST, followed by incubation with secondary antibody for at least 2 h. Finally, samples were washed with PBST, stained for 20 min with 1 μg ml$^{-1}$ DAPI in PBST, before washing once more with PBST and mounting of the disks in Vectashield mounting medium (Vector Labs). Samples were stored at 4 °C until imaging. Confocal images were acquired on a Zeiss LSM510 laser scanning microscope using a water immersion ×40 objective. To visualize apical surfaces (where Expanded protein is localized), Z-stacks of between 8 and $20 \times 1$ μm

sections were collected, and then processed as maximum intensity projections. Images were processed using ImageJ, and figures prepared using Adobe software.

**Western blotting and PCR genotyping of transgenic flies.** To analyze adult flies using immunoblotting, three flies per genotype were frozen at −80 °C overnight, followed by homogenization on ice in 20 μl of Laemmli sample buffer per fly (1610737, Bio-Rad,) using a Squisher manual homogenizer (Zymo Research). After boiling samples at 95 °C for 10 min, samples were centrifuged and supernatants transferred to fresh tubes. Finally, DTT was added to a final concentration of 50 mM and equal volumes of samples were analyzed by SDS-PAGE, followed by western blotting. For PCR genotyping, flies were collected, frozen in liquid nitrogen, and subsequently crushed using a micropestle, followed by preparation of genomic DNA using a Qiagen kit (69506). Genomic DNA was stored at − 20 °C, followed by PCR genotyping using the following primers: 5′-GCCGCTCAAGA-TAGCCAGAT-3′ and 5′-GCACACTTCCTGGAACCGCTCGCATC-3′ to detect the Roo transposon; and 5′-AGAGGCGCTTCGTCTACGGAGCGACA-3′ and 5′-CGGCAAGCAGGCATCGCCATGGGTC-3′ to detect the FRT site.

**Statistical analysis.** Graphics and statistical analyses were carried out using the GraphPad Prism software. Data are presented as mean ± s.e.m., unless stated otherwise. The significance of differences between the means or the population distributions was determined using two-tailed unpaired Student *t*-test. Differences were considered statistically significant when *p*-values were below 0.05 (*), 0.01 (**), or 0.001 (***). *p*-values are listed in the corresponding figure legends where appropriate.

**Data availability.** The coordinates and structure factor files of the complex of human MOB1 (residues 33–216) bound to NDR2 (residues 25–88) have been deposited in the Protein Data Bank (PDB) with accession number 5XQZ. All relevant data are available from the corresponding authors upon reasonable request.

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

## Acknowledgements

We thank J. Lisztwan, P. Rodriguez-Viciana. P. Ribeiro and S. Ultanir for critical reading of the manuscript, and all members of our laboratories for helpful feedback. We thank the staff at the BL17U1 beamline at the Shanghai Synchrotron Radiation Facility. The pAW vector, Mats, and Trc cDNAs were kindly provided by the Drosophila Genomics Resource Center (Indiana University, USA), which is supported by the NIH grant 2P40OD010949-10A1. The pLEX, pMD.G, p8.91 and plasmids were generously provided by P. Rodriguez-Viciana (UCL Cancer Institute). Anti-Expanded antibody was kindly provided by A. Laughon (University of Wisconsin–Madison, USA). The Hergovich laboratory was supported by Wellcome Trust (090090/Z/09/Z), BBSRC (BB/I021248/1), the National Institute for Health Research University College London Hospitals Bio-medical Research Centre, and UCL Cancer Research UK Centre funding. Y.K. was sponsored by the Ministry of National Education (The Republic of Turkey). G.W. was funded by grants from the National Natural Science Foundation of China (grants 31470223 and 31670106), State Key Laboratory of Microbial Resources, Institute of Microbiology, Chinese Academy of Sciences, and the Program for Professor of Special Appointment (Eastern Scholar) at Shanghai Institutions of Higher Learning. The Tapon lab was supported by the Francis Crick Institute, which receives its core funding from Cancer Research UK (FC001175), the UK Medical Research Council (FC001175), and the Wellcome Trust (FC001175). The Bjedov lab was supported by a European Research Council Starting Grant (311331) and UCL Cancer Research UK Centre funding.

## Author contributions

Y.K. performed and interpreted most experiments in Figs. 3–6 and the Supplementary Figures. K.L., supported by Z.G., performed and interpreted most experiments in Figs. 1 and 2 and Supplementary Figs. 2–4, 10, 11, and 18. M.H. and Y.K. supported by C.L. performed the experiments described in Figs. 5, 6, and Supplementary Fig. 16. M.G., B.A.S., M.M., L.H., A.A.D.S. and E.S.J.S. assisted Y.K. with the experiments shown in Supplementary Figs. 5–15. I.B. and N.T. participated in the analyses and interpretation of all experiments, in particular the design, analyses, and interpretation of Figs. 5, 6 and

Supplementary Fig. 16. G.W. and A.H. participated in the conception, planning, analyses, and interpretation of all experiments, and wrote the manuscript. All authors approved the submitted manuscript version.

## Additional information

**Competing interests:** The authors declare no competing financial interests.

