## [Peer Review File · Nature Communications]

Reviewers' Comments:

Reviewer #1 (Remarks to the Author)

Kulaberoglu et al report the structure of a complex between N-truncated MOB1 and the NTR domain of NDR2. Comparing with the previously reported structures, they found mutations differentially affecting binding to target kinases, MST (Hippo), LATS (Warts), and NDR (Trc) and then performed binding experiments and in vivo assays using the mutant MOB1. This report contains voluminous experiments, which should be appreciated, but the results contains little new findings. Based on the obtained structure, the authors failed to produce a mutation that discriminate LATS/Hippo and NDR/Trc kinases. With other mutants, the authors discussed a "stable" MOB1-Hpo binding, whereas this binding seems to have no impact on the signaling since MST/Hpo phosphorylates MOB1 without this binding. Overall, this report fails to provide novelty enough for publication in this journal.

Reviewer #2 (Remarks to the Author)

The authors manuscript concerns the differential binding of Mob to the Ndr/Lats and Hippo kinases. There identify structural determinants for the interactions and locate key residues. Mutation of these residues give selectivity for interaction partners and allow the biological dissection for the Hippo-Mob from the Mob-wts interaction. Overall the data is convincing and thorough, giving much needed insight into regulation of the Hippo pathway. I do have some concerns that I feel the authors could address to be complete.

Major

What happens to association with Ndr1 and Lats2 association with the mutant Mob?

The biological role of Ndr is inferred to be less important then Lats. A mob1 mutant that selectively binds Lats over Ndr should be attempted in the developmental setting to claim that the mob1-wts pathway is important. mob1-trc could have the same effect therefore altering the overall conclusion of the paper.

In supplementary figures S4E, S5E and S6E panels are not labelled. The E51K/K104E/K105E construct isnt labelled on this figure but is included in figure 3C, I would assume this is the unlabelled well, but it is negative for anti-HA and anti-Myc. Either way they need to include their E51K/K104E/K105E blots for all of the Drosophila IPs as they are not included.

Minor

For figure 4 and supplementary fig 10 better images are required

Supplementary figure S5D. K104E/K104E when it should be K104E/K105E. This is the same blot as S5E but shows less constructs.

They say their processed "Diffraction data 2.10 Å". I think they mean the data had a resolution of this. If this is the resolution then its somewhat low as they used a high-resolution beamline

Reviewer #3 (Remarks to the Author)

The Hippo signaling pathway is essential to both development and control of tissue growth. However, we lack mechanistic understanding of how the core Hippo components function in a physiological situation, specifically what interactions are relevant in vivo, and how do these interactions modulate the effector kinase activity.

The three core components of interest here are a STE20 kinase (Mst1/2 in mammals and Hippo (Hpo) in *Drosophila*), the Mob protein adaptor (Mats in *Drosophila*) and the effector kinase of the NDR kinases family (Lats1/2 and NDR1/2 in mammals and Warts and Tricornered in *Drosophila*). NDR kinases must be bound by Mob proteins and phosphorylated by Hpo kinases to become catalytically active. Mobs are also bound and phosphorylated by Mst1/2 kinases *in vitro*, an event that increases Mob binding affinity for NDR kinases. However, the physiological role of this interaction is uncertain. For example, Vrabioiu and Struhl show in the context of *Drosophila* wing growth that Hpo, and therefore Hpo binding and phosphorylation of Mats, is not required for Mats to trigger a conformation change in Warts that is essential for Warts activity. Similarly, Ni et al show that Mob phosphorylation by Mst2 is not required for Lats1 activation in HEK293 cells. Instead, they propose that Mob binding to Mst2 primes Mob for Lats1 activation, whereas Mob phosphorylation by Mst2 triggers complex disassembly.

Thus, a major challenge is to determine the role of the Mob/Mst interaction in controlling activity of the Mob/NDR/Mst complex both *in vivo* and *in vitro*, and in particular, to determine the potentially distinct roles of Mst2/Mob binding versus phosphorylation of Mob by Mst2. In the present manuscript, the authors determine the crystal structure of human Mob1 bound to the NDR2 Mob binding domain, and use insights achieved from this structure to design and test mutant Mob variants with selective loss of interaction with Lats1/2 and Mst1/2.

Both the crystal structure as well as the identification of Mob mutations that selectively reduce or abolish binding interactions with either Mst1/2 or Lats1/2 kinases is a significant advance in the field, and on this basis, I would argue for publication of the findings. However, there are significant deficiencies/discrepancies with previous findings in the *in vitro* data set that I think should be resolved and discussed explicitly before the paper is accepted. The validity of the physiological conclusions depends on that. Accordingly, I cannot recommend publication of the paper in its present form. However, the authors have generated a great deal of valuable data, new approaches and assays. Hence, I would recommend publication in *Nature Communications* if the main issues, below, can be addressed.

Main issues

1. The authors incorrectly assume that Mob phosphorylation by Mst2 is required for Lats1 activity *in vivo*. This issue is not resolved in the field, as pointed out in the introduction and should be made clear throughout the manuscript. The authors' assumption stems from their inability to detect an *in vitro* interaction between Mob and Lats1 in the absence of Mob phosphorylation. The description of this result is not consistent throughout the Results' section and the Discussion. It ranges from "we detect no binding using our assay" (correct) to "phosphorylation is essential for Mob1 binding to Lats1". The latter statement is incorrect, or at least it conflicts with previous *in vitro* accounts (Ni et al, 2015 and Kim et al, 2016) that document binding between un-phosphorylated Mob and Lats1 using different assays and, in addition, it has no *in vivo* bearing. The authors should discuss possible reasons for the discrepancy. One obvious starting point is that their assay, isothermal titration calorimetry (ITC), may require higher concentration of reagents in order to detect a lower affinity interaction, or that other strategies such as competition/displacement ITC assays with higher affinity ligands may be required.

Therefore the authors should stress the relevant results, which are the relative affinities of the different interaction pairs that they nicely measure, rather than make absolute, general statements with *in vivo* implications.

2. The authors rest the case of proving the importance of the *in vivo* Mob-Mst1/2 interaction on a single Mob mutant variant. Therefore it is essential that their Mob variant is thoroughly characterized *in vitro*, and possible discrepancies are acknowledged/ discussed and addressed experimentally.

Ni et al, 2015 described a crystal structure of Mob1 bound to an Mst2 peptide, characterized the interaction to ~100nM affinity using ITC, and identified two binding sites on the Mob protein essential for Mst2 binding. Based on the crystal structure they produced Mob variants with Mst2 binding reduced to various degrees. They observed a correlation between the variant's ability to

bind Mst2 and the ability of Mst2 to phosphorylate the Mob variant. They concluded that Mob's ability to be phosphorylated by Mst2 is tightly linked to its ability to bind Mst2.

The authors of this manuscript produce their own Mob variant deficient in Mst2 binding by screening Mob1 variants with substitutions of charged, surface exposed amino acid residues in a low stringency co-immunoprecipitation assay. They identify two lysine residues (K104, and K105) as essential for stably binding to Mst1/2. The Mob variant they identify (MobK104EK105E) can however be efficiently phosphorylated in their kinase assay. They infer that Mst2 and MobK104EK105E interact transiently (via a low affinity interaction) and that that interaction is sufficient for their Mob variant to be phosphorylated.

My concern is that the authors' variant retains more Mst2 binding activity than it appears in their assay, and therefore their *in vivo* conclusions are meaningless.

The two accounts use different assays to assess binding and phosphorylation. Thus, at a minimum the authors should do a side by side comparison between the abilities of their variant and one of the Ni et al variants (MobKRR3A, for example) to bind Mst2 and be phosphorylated by Mst2. It is important that they use assays with appropriate dynamic range. For that, the Ni et al assays are good starting points as they are able to quantitate differences between the Mob variants. Possible sources for the discrepancy between this manuscript and the Ni et al manuscript could include:

- Differences in the ionic strength of the buffers used for co-immunoprecipitations between the two accounts. The low stringency buffer used by the manuscript under review is a buffer with sub physiological ionic strength; however that may not be low stringency for the Mob-Mst2 interaction, which could be driven by hydrophobic interactions.

- Differences in the buffer and stoichiometry conditions of the kinase assay. The authors of this manuscript use no salt in their kinase buffer, and much higher relative amounts of kinase.

Other experiments that would strengthen and broaden the scope of this account would be to test the Ni et al variants mentioned above (deficient in both Mst2 binding, and phosphorylation by Mst2) as well as the un-phosphorylatable Mob variants (Mob2TA, Ni et al) in the physiological assays developed by the current study.

In addition, the authors could devise a binding assay in a close to physiological setting. For example, in the Fig. 4 experiments, they show that both wild type Mob and the MobK104EK105E variant suppress MCF-7 cells proliferation. Can the Mob variant bind endogenous Mst2 in this situation where it is clearly functionally active?

Another concern about the MobK104EK105E is that it has increased affinity for Lats1 and that the increased affinity is sufficient to bypass an Mst2 binding requirement in *in vivo* settings.

Two recent papers (Ni et al, 2015 and Kim et al, 2016) show that Mobs have an auto-inhibited state, and that binding and/or phosphorylation by Mst2 relieves that inhibition by increasing Mob's affinity for NDR kinases. I am concerned that the MobK104EK105E used in this studies, as selectively abolishing Mob-Mst2 stable interactions, has altered affinity for NDR kinases, in particular an increased affinity that could compensate for the loss of Mst2 binding. To address this concern, the authors should compare quantitatively the interactions (p)Mob1 K104E K105E - Lats1/NDR2 and (p)Mob1 -Lats1/NDR2.

3. The "updated" NDR kinase activation model proposed in the Discussion is not consistent with available data and does not follow from the authors' results.

The authors should specifically discuss their contribution to the mechanistic picture of NDR kinase activation. Their results suggest that stable Mob binding to Lats1/2, but not to Mst1, is required for Lats1/2 activation in physiological settings. This argues against models that involve ternary complex formation (Ni et al, 2015), but as they are (not addressing the role of phosphorylation) they produce no new information towards understanding how Mob phosphorylation fits into the *in vivo*, physiological picture (which the authors depict as an established step in Fig. S12). The authors should acknowledge the difficulty of reconciling all data in a general model. Indeed Vrabioiu and Struhl results show that Hpo is not required for Mats binding to Warts and triggering

an activating conformational change. Thus in the particular context of the *Drosophila* wing growth Hpo (and thus Hpo binding and Hpo phosphorylation) is not required for Mats-Lats1 binding, although Mats phosphorylation by a different kinase may be required. The authors' model (Fig. S12) could exemplify a mechanism for a switch like activation of NDR kinases in specific biological contexts (for example, a situation when a growth response should be abruptly terminated) and can be presented while acknowledging its limitations.

The authors should discuss the Ni et al mutants and suggest possible explanations for why their variant behaves differently than the Ni et al variants.

Minor points

- Line 91 of the manuscript claims Mob1 variants with selective NDR1/2 impairment, but those variants were not described here (Mob1 E51K did not pass the criteria for selective impairment).
- The statements in lines 128, 178-179, and 229-231 are superfluous and do not say much in the context of the specific results.
- Line 472, what is assembly buffer?

References

Ni L, Zheng Y, Hara M, Pan D, Luo X. Structural basis for Mob1-dependent activation of the core Mst-Lats kinase cascade in Hippo signaling. *Genes & development* 29, 1416-1431 (2015).

Vrabioiu AM, Struhl G. Fat/Dachsous Signaling Promotes *Drosophila* Wing Growth by Regulating the Conformational State of the NDR Kinase Warts. *Dev Cell* 35, 737-749 (2015).

Kim SY, Tachioka Y, Mori T, Hakoshima T. Structural basis for autoinhibition and its relief of MOB1 in the Hippo pathway. *Sci Rep* 6, 28488 (2016).

Reviewers' comments

Reviewer #1 (Remarks to the Author):

Kulaberoglu et al report the structure of a complex between N-truncated MOB1 and the NTR domain of NDR2. Comparing with the previously reported structures, they found mutations differentially affecting binding to target kinases, MST (Hippo), LATS (Warts), and NDR (Trc) and then performed binding experiments and in vivo assays using the mutant MOB1. This report contains voluminous experiments, which should be appreciated, but the results contains little new findings. Based on the obtained structure, the authors failed to produce a mutation that discriminate LATS/Hippo and NDR/Trc kinases. With other mutants, the authors discussed a “stable” MOB1-Hippo binding, whereas this binding seems to have no impact on the signaling since MST/Hippo phosphorylates MOB1 without this binding.

Overall, this report fails to provide novelty enough for publication in this journal.

RESPONSE:

First of all, we would like to thank reviewer #1 for stating that our study contains voluminous experiments that should be appreciated. However, we are quite surprised by the sentences that follow the initial summary statement. Since we do not agree with reviewer #1 about their assessment of our work, we are first directly responding to these sentences:

1) Our MOB1 mutants clearly discriminate between Hippo core kinases

As is clearly evident from our data, the D63V mutant only displays selective loss-of-interaction with human LATS1/2 and fly Warts, while binding to human MST1/2, fly Hippo, human NDR1/2 and fly Trc remains intact (for examples see Figure 3 and suppl. Figures S5 to S7 in our revised manuscript). The K104E/K105E mutant only affects binding to human MST1/2 and fly Hippo, while the interactions with human LATS1/2, fly Warts, human NDR1/2 and fly Trc remain intact (for examples see Figure 3 and suppl. Figures S5 to S7). Furthermore, the D63V/K104E/K105E mutant still binds to human NDR1/2 and fly Trc, but not to human LATS1/2, fly Warts, human MST1/2 and fly Hippo (for examples see Figure 3 and suppl. Figures S5 to S7). Therefore, we have successfully generated MOB1 mutants that clearly discriminate between the human Hippo core kinases MST1/2, LATS1/2 and NDR1/2 (and fly Hippo, Warts and Trc, respectively).

2) Our data suggest that a stable Hippo/MOB1 (MST1/2-MOB1) complex is non-essential

Yes, as rightly observed by reviewer #1, our data indicate that a stable complex between MOB1 and fly Hippo (human MST1/2) is not required in the context of the biological parameters that were tested in our study. Of course, we cannot rule out the role of very short transient interactions, given that MOB1 serves as a substrate of Hippo (MST1/2). Nevertheless, considering the recent strong emphasis on the importance of a stable MOB1/MST2 complex in Hippo signaling (for examples see Ni et al. (2015) and Meng et al. (2016)), our study is very important in order to provide significant insights into the biological importance of such a stable complex (at least in the biological contexts that we tested).

Overall as listed below, our study provides several new mechanistic as well as biological insights into the role(s) of MOB1 as a master regulator of Hippo core signaling. In particular, we would like to stress that our study defines the following advances of significance in Hippo core signaling:

- (i) We report the first crystal structure of the MOB1/NDR2 complex, thereby laying the foundation for comparative studies of MOB1/NDR vs. MOB1/LATS protein complexes.
- (ii) Using a multi-disciplinary approach we discovered that distinct MOB1 residues mediate the interactions of MOB1 with the different Hippo core kinases in mammalian and fly cells, thereby enabling MOB1 to regulate the specificity and amplitude of Hippo core signaling.
- (iii) These new structural and molecular insights into the Hippo core cassette provide notable insights into how MOB1 differentiates between Hippo core kinase, thereby defining a

framework for how these different interactions may function in different cellular contexts and establishing a significant foundation to dissect the biological importance of the regulatory interactions of MOB1 with Hippo core kinases.

- (iv) Based on our structural and biochemical discoveries, we could design completely novel MOB1 variants with selective loss-of-interaction, thereby empowering us to discover in human cancer cells and transgenic flies that selected regulatory interactions of MOB1 are essential for development, tissue growth control and Yki regulation, hence establishing MOB1 as a central signal transducer hub of the Hippo core network.
- (v) Overall we established key research tools to probe the significance of the regulatory interactions of MOB1, thereby empowering the translation of our structural and biochemical data into studies of human cells and flies (see point (iv) above) and future studies of MOB1-regulated Hippo signaling in diverse disease-relevant settings.
- (vi) Our biochemical and genetic data help to settle the controversy regarding the role of a stable MOB1/Hippo interaction.
- (vii) By expanding the understanding of the regulatory MOB1 interactions in the Hippo core cassette, we expand the understanding of the activation mechanisms of NDR/LATS kinases.

Reviewer #2 (Remarks to the Author):

The authors manuscript concerns the differential binding of Mob to the Ndr/Lats and Hippo kinases. There identify structural determinants for the interactions and locate key residues. Mutation of these residues give selectivity for interaction partners and allow the biological dissection for the Hippo-Mob from the Mob-wts interaction. Overall the data is convincing and thorough, giving much needed insight into regulation of the Hippo pathway. I do have some concerns that I feel the authors could address to be complete.

RESPONSE:

We would like to thank reviewer #2 for their clear support and recognition of the importance and impact of our findings. We also believe that our study provides much needed insights into the regulation of the Hippo pathway. Equally important, we are convinced that our study will stimulate several novel lines of research in the future. Furthermore, we are also confident that we have sufficiently addressed the concerns of reviewer #2 in our revised manuscript and responses below.

Major

- 1) *What happens to association with Ndr1 and Lats2 association with the mutant Mob?*

RESPONSE:

Our revised manuscript covers the testing of human MST1, MST2, LATS1, LATS2, NDR1 and NDR2 as well as fly Hippo, Warts and Trc. In particular, we would like to draw the reviewer's attention to:

-) Figure 3 and suppl. Figures S5 to S13 in our revised manuscript

The structural and biochemical characterization of the three main mutants D63V, K104E/K105E and D63V/K104E/K105E is shown in these main and supplementary figures. A summary of our extensive characterization efforts is presented in Figure 3c.

-) new suppl. Figures S3 and S11 (to complement Figure 2c-f)

To expand our biochemical characterization, we are now including new supplementary data, covering ITC assays testing the binding of full-length MOB1 wild-type and K104E/K105E to the wild-type and mutant NTR domains of human NDR1, NDR2, LATS1 and LATS2.

Thus, we are now including in the revised manuscript a full coverage of MOB1 interactions with human NDR1, NDR2, LATS1 and LATS2 using ITC and co-immunoprecipitation approaches.

As is evident from these data sets, the D63V mutant only displays selective loss-of-interaction with human LATS1/2 and fly Warts, while binding to human MST1/2, fly Hippo, human NDR1/2 and fly Trc remain intact. The K104E/K105E mutant only affects binding to human MST1/2 and fly Hippo, while the interactions with human LATS1/2, fly Warts, human NDR1/2 and fly Trc remain intact. The D63V/K104E/K105E mutant still binds to human NDR1/2 and fly Trc, but not to human LATS1/2, fly Warts, human MST1/2 and fly Hippo.

- 2) *The biological role of Ndr is inferred to be less important than Lats. A mob1 mutant that selectively binds Lats over Ndr should be attempted in the developmental setting to claim that the mob1-wts pathway is important. mob1-trc could have the same effect therefore altering the overall conclusion of the paper.*

RESPONSE:

We fully agree that it would be wonderful to also test a MOB1 mutant with selective loss-of-interaction regarding NDR1/2 (a MOB1 mutant with intact binding to human MST1/2, fly Hippo, human LATS1/2 and fly Warts, but loss of stable interaction with human NDR1/2 and fly Trc). Actually, we would love to have such a MOB1 mutant based on the central interest of our lab on NDR1/2 signaling, but unfortunately we currently do not see a way to pursue

this avenue experimentally. More specifically, over the past years neither our structure-based approaches (see Figures 1 and 2 and suppl. Figure S2) nor the mutagenesis of charged residues on the surface of MOB1 followed by interactions studies (see Figure 3 and suppl. Figures S5 to S7) resulted in the generation of a MOB1 mutant that solely displays loss of NDR1/2 (Trc) binding without severely impacting MST1/2 (Hippo) and/or LATS1/2 (Warts) binding. Overall, based on the available crystal structures it appears that LATS1/2 (Warts) kinases have additional contact sites with MOB1 when compared to NDR1/2 (Trc) binding to MOB1 (see Figures 1 and 2 and suppl. Figure S2, as well as the figure legends corresponding to these figures). All bonds between NDR2 and MOB1 are also fully present in the LATS1/MOB1 interactions (see Figures 1 and 2 and suppl. Figure S2, as well as the figure legends corresponding to these figures). Thus, we believe that it is most likely impossible to design a much desired MOB1 mutant with selective loss-of-interaction of NDR1/2 (at least based on our current structural understanding of these direct interactions).

- 3) *In supplementary figures S4E, S5E and S6E panels are not labelled. The E51K/K104E/K105E construct isnt labelled on this figure but is included in figure 3C, I would assume this is the unlabelled well, but it is negative for anti-HA and anti-Myc. Either way they need to include their E51K/K104E/K105E blots for all of the Drosophila IPs as they are not included.*

RESPONSE:

In panels S4e, S5e and S6e we only showed lane 6 to include a further negative control (additional demonstration of specificity of our co-IPs). In all three panels, lane 6 represents untransfected control lysates that underwent the same procedures as the other transfected samples, hence they are negative for anti-HA and anti-myc as expected for an untransfected negative control. We are sorry for not labelling these lanes properly, which we have now corrected in the revised manuscript version (please note that Figures S4, S5 and S6 from the first manuscript version now correspond to Figures S5, S6 and S7 in the revised version).

Furthermore, we feel that our manuscript is already at risk for “information overload” regarding MOB1 mutants, hence we do not include E51 data in detail, but rather provide a comprehensive summary in Figure 3c. However, since the E51-based mutants are not essential at all for the central messages of our manuscript, we are fully prepared to remove any text and information relating to E51 from the manuscript, if the reviewer prefers (i.e. reformat Figure 3c and the corresponding main text accordingly).

Minor

- A) *For figure 4 and supplementary fig 10 better images are required.*

RESPONSE:

As requested, higher resolution images have been included for Figures 4c, 4e and S10a. Please note that Figure S10 now corresponds to Figure S14 in the revised version.

- B) *Supplementary figure S5D. K104E/K104E when it should be K104E/K105E. This is the same blot as S5E but shows less constructs.*

RESPONSE:

The labelling of Figure S5 (now labelled as Figure S6 in the revised manuscript) has been corrected. Thank you very much for noticing the labelling mistake.

- C) *They say their processed "Diffraction data 2.10 Å". I think they mean the data had a resolution of this. If this is the resolution then its somewhat low as they used a high-resolution beamline.*

RESPONSE:

The crystal structure of the NDR2/MOB1 complex was resolved at 2.1 Å (as indicated in Table S1) for the following reasons:

First, to preserve only the high quality diffraction data, we processed our data in a conserved manner, hence the completeness, signal-to-noise ratio I/σ_I , and redundancy values for the highest resolution shell of our diffraction data were 100%, 3.1, and 6.4, respectively. We could have pushed for a resolution of 2.0 Å or below 2.0 Å, but this would have compromised data quality. For example, the signal-to-noise ratio, I/σ_I , for the highest resolution shell would drop to around 2, or even closer to 1; and the completeness for the highest resolution would drop to lower than 95%, or even close to 90%. Collectively, this unnecessary decrease of values would compromise the certainty of our electron density map and structure model.

Second, the resolution of a protein structure depends on the particular protein/protein complex under study, and it is not uncommon that structures of protein complexes have medium or low resolution although the beamline used for collecting diffraction data displays high-intensity X-ray beams. In this regard, we already reported over the past years several protein/protein complexes with 2.10 Å or lower resolution based on data collected at the same beamline, BL17U1 of SSRF. Please see a list of recent examples below:

Zhang Z, Chen L, Gao L, Lin K, Zhu L, Lu Y, Shi X, Gao Y, Zhou J, Xu P, Zhang J, Wu G. Structural basis for the recognition of Asef by Adenomatous Polyposis Coli. *Cell Res.* 22:372-386 (2012)

Note: APC (407-751) in complex with Asef (170-194) resolved at 2.30 Å.

Zhang Y, Fu L, Qi X, Zhang Z, Xia Y, Jia J, Jiang J, Zhao Y, Wu G. Structural insight into the mutual recognition and regulation between Suppressor of Fused and Gli/Ci. *Nat. Commun.* 4:2608 (2013)

Note: human Sufu Δ 60, Sufu Δ 20 and full-length at 2.25 Å, 3.10 Å and 3.20 Å, respectively.

Zhang Z, Akyildiz S, Xiao Y, Gai Z, An Y, Behrens J, Wu G. Structures of the APC-ARM domain in complexes with discrete Amer1/WTX fragments reveal that it uses a consensus mode to recognize its binding partners. *Cell Discov.* 1:15016. (2015)

Note: APC-ARM in complex with Amer1-A2 at a resolution of 2.10 Å.

Gai Z, Chu W, Deng W, Li W, Li H, He A, Nellist M, Wu G. Structure of the TBC1D7-TSC1 complex reveals that TBC1D7 stabilizes dimerization of the TSC1 C-terminal coiled coil region. *J. Mol. Cell. Biol.* 8:411-425. (2016)

Note: TBC1D7 in complex with TSC1 (939-992) at a resolution of 2.80 Å.

Gai Z, Wang Q, Yang C, Wang L, Deng W, Wu G. Structural mechanism for the arginine sensing and regulation of CASTOR1 in the mTORC1 signaling pathway. *Cell Discov.* 2:16051. (2016b)

Note: CASTOR1 in complex with arginine at a resolution of 2.07 Å.

Reviewer #3 (Remarks to the Author):

The Hippo signaling pathway is essential to both development and control of tissue growth. However, we lack mechanistic understanding of how the core Hippo components function in a physiological situation, specifically what interactions are relevant in vivo, and how do these interactions modulate the effector kinase activity.

The three core components of interest here are a STE20 kinase (Mst1/2 in mammals and Hippo (Hpo) in Drosophila), the Mob protein adaptor (Mats in Drosophila) and the effector kinase of the NDR kinases family (Lats1/2 and NDR1/2 in mammals and Warts and Tricornered in Drosophila). NDR kinases must be bound by Mob proteins and phosphorylated by Hpo kinases to become catalytically active. Mobs are also bound and phosphorylated by Mst1/2 kinases in vitro, an event that increases Mob binding affinity for NDR kinases. However, the physiological role of this interaction is uncertain. For example, Vrabioiu and Struhl show in the context of Drosophila wing growth that Hpo, and therefore Hpo binding and phosphorylation of Mats, is not required for Mats to trigger a conformation change in Warts that is essential for Warts activity. Similarly, Ni et al show that Mob phosphorylation by Mst2 is not required for Lats1 activation in HEK293 cells. Instead, they propose that Mob binding to Mst2 primes Mob for Lats1 activation, whereas Mob phosphorylation by Mst2 triggers complex disassembly.

Thus, a major challenge is to determine the role of the Mob/Mst interaction in controlling activity of the Mob/NDR/Mst complex both in vivo and in vitro, and in particular, to determine the potentially distinct roles of Mst2/Mob binding versus phosphorylation of Mob by Mst2. In the present manuscript, the authors determine the crystal structure of human Mob1 bound to the NDR2 Mob binding domain, and use insights achieved from this structure to design and test mutant Mob variants with selective loss of interaction with Lats1/2 and Mst1/2.

Both the crystal structure as well as the identification of Mob mutations that selectively reduce or abolish binding interactions with either Mst1/2 or Lats1/2 kinases is a significant advance in the field, and on this basis, I would argue for publication of the findings. However, there are significant deficiencies/discrepancies with previous findings in the in vitro data set that I think should be resolved and discussed explicitly before the paper is accepted. The validity of the physiological conclusions depends on that. Accordingly, I cannot recommend publication of the paper in its present form. However, the authors have generated a great deal of valuable data, new approaches and assays. Hence, I would recommend publication in Nature Communications if the main issues, below, can be addressed.

RESPONSE:

We are very thankful for the positive and clearly supportive response of reviewer #3. As outlined below, we have addressed in detail the main and minor concerns raised by reviewer #3. Overall, we are confident that we have sufficiently addressed the reviewer's concerns in order to justify publication in *Nature Communication*.

Main issues

1. The authors incorrectly assume that Mob phosphorylation by Mst2 is required for Lats1 activity in in vivo. This issue is not resolved in the field, as pointed out in the introduction and should be made clear throughout the manuscript. The authors' assumption stems from their inability to detect an in vitro interaction between Mob and Lats1 in the absence of Mob phosphorylation. The description of this result is not consistent throughout the Results' section and the Discussion. It ranges from "we detect no binding using our assay" (correct) to "phosphorylation is essential for Mob1 binding to Lats1". The latter statement is incorrect, or at least it conflicts with previous in vitro accounts (Ni et al, 2015 and Kim et al, 2016) that document binding between unphosphorylated Mob and Lats1 using different assays and, in addition, it has no in vivo bearing. The authors should discuss possible reasons for the discrepancy. One obvious starting point is that their assay, isothermal titration calorimetry (ITC), may require higher concentration of reagents in order to detect a lower affinity interaction, or that other strategies such as competition/displacement ITC assays with higher affinity ligands may be required.

Therefore the authors should stress the relevant results, which are the relative affinities of the different interaction pairs that they nicely measure, rather than make absolute, general statements with in vivo implications.

We agree with the reviewer that we need to stress the differences between *in vitro* and *in vivo* evidence, as well as discuss our ITC results in the context of the published works by Ni et al. (2015) and Kim et al. (2016). Therefore, we have re-phrased/re-written/expanded our manuscript on pages 4, 8, 9, 10 and 20 (highlighted in yellow in the revised manuscript).

2. The authors rest the case of proving the importance of the *in vivo* Mob-Mst1/2 interaction on a single Mob mutant variant. Therefore it is essential that their Mob variant is thoroughly characterized *in vitro*, and possible discrepancies are acknowledged/ discussed and addressed experimentally.

RESPONSE:

We fully agree with reviewer #3 that the MOB1 K104E/K105 mutant needs to be very properly characterized *in vitro* in order to support our *in vivo* conclusions. Therefore, we have spent quite some time to thoroughly characterize our K104E/K105E mutant using different approaches (for more detailed discussions see our responses below).

Please see the following 9 figures showing a thorough *in vitro* characterization of K104E/K105E:

- Figure 3
- Figure S5
- Figure S6
- Figure S7
- Figure S8
- Figure S10
- Figure S11
- Figure S13
- Figure S15

We would have loved to test a second MOB1 mutant that is specifically deficient in binding to full-length wild-type MST1/2 (Hippo), but as outlined in our responses below, Ni et al. (2015) did not experimentally define such a mutant in their study. Moreover, our testing of MOB1 K153/R154 mutants showed that the K153/R154 residues are required for LATS2, Warts, MST2 and Hippo binding in human and fly cells, respectively (see new suppl. Fig. S9).

Moreover, we would like to stress that every study of mutants has its limitations. For example, the effect of studying a specific phospho-acceptor site is limited to the manipulations of the respective phosphorylated residue (i.e. phospho-acceptor vs. phospho-mimicking).

Ni et al, 2015 described a crystal structure of Mob1 bound to an Mst2 peptide, characterized the interaction to ~100nM affinity using ITC, and identified two binding sites on the Mob protein essential for Mst2 binding.

RESPONSE:

Ni et al. (2015) reported the crystal structure of a Thr378-phosphorylated MST2 peptide (encompassing residues 371 to 400 of MST2) bound to MOB1 lacking the first 50 amino acids (MOB1 Δ 50) at a resolution of 2.65 Å. Moreover, Ni et al. (2015) reported that full-length wild-type MOB1 bound *in vitro* to a Thr378-phosphorylated MST2 peptide (encompassing residues 371 to 400 of MST2) with an affinity of 119nM based on ITC in a standard buffer system (20mM Tris pH8.0, 100mM NaCl, and 2mM MgCl₂). Based on the structure of the 30aa fragment of MST2 (371 to 400) bound to MOB1, Ni et al. (2015) could describe two MOB1/MST2 interfaces. A main pT-binding site, centering around phosphorylated Thr378 of MST2 and residues K153, R154 and R157 of MOB1, as well as a supportive HS binding site (residues 390 to 398 of MST2), which makes hydrophobic interactions with MOB1.

Noteworthy, this MOB1/MST2 crystal structure is fully focused on one selected region of MST2 (residues 371 to 400 only) with phosphorylated Thr378 as a central interaction site. Other important sites such as Thr349, Thr356, Thr364 on MST2 (see Figure 3D in Ni et al., 2015) are not covered by this crystal structure. Therefore, it is not surprising that the published MOB1/MST2 crystal structure (see Ni et al., 2015) does not cover Lys104 and Lys105 of MOB1. We actually believe that it is quite possible that the positively charged Lys104/Lys105 residues of MOB1 may bond with the negatively charged phosphorylated Thr residues at 349, 356 and/or 364 or possibly even other residues on MST2.

Overall, it is possible that Ni et al. did not identify the main interaction site on MOB1 with respect to MST2 binding based on their focused approach on residues 371 to 400 of MST2. This notion is supported by two recent publications by the Sicheri and Gingras laboratories (see Couzens et al., 2017 and Xiong et al., 2017). In a nutshell, these two recent studies provide structural evidence suggesting that at least six phospho-threonine sites (T329, T340, T353, T367, T380 and T387) and a seventh hydrophobic non-phospho site on human MST1 contribute to MOB1 binding. Interestingly, their data suggest that Pro106 significantly contributes to phospho-threonine binding on MST1.

We have expanded our main text on pages 17 and 18 to properly discuss these points and publications in the context of our findings.

Based on the crystal structure they produced Mob variants with Mst2 binding reduced to various degrees. They observed a correlation between the variant's ability to bind Mst2 and the ability of Mst2 to phosphorylate the Mob variant. They concluded that Mob's ability to be phosphorylated by Mst2 is tightly linked to its ability to bind Mst2.

RESPONSE:

In Figure 2D and Suppl. Figure 3D of their study, Ni et al. (2015) compared the interactions of an unphosphorylated MST2 fragment (371 to 427) vs. a phosphorylated MST2 fragment (371 to 427) with wild-type full-length MOB1. This biochemical approach (GST pull-downs) confirmed the importance of the K153, R154 and R157 residues of MOB1 regarding to binding to the pThr-binding site on MST2, as predicted by the MOB1/MST2 crystal structure (Ni et al., 2015).

However, Ni et al. did not study the interaction of full-length wild-type MST1/2 (or Hippo) with full-length MOB1 (or Mats) mutants. In other words, Ni et al. (2015) did not produce full-length MOB1 variants for which they experimentally demonstrated that they are indeed defective in MST2 (or MST1 or Hippo) binding when the interactions of full-length proteins were assessed. In general, we would like to stress that the Ni et al. (2015) study has to be assessed from the right perspective. Regarding the MST2/MOB1 interaction, Ni et al. clearly focused on defining which regions/residues of MST2 are required for MOB1 binding in a full-length context (i.e. see Figure 3 and suppl. Figure S4 in Ni et al., 2015). In this regard, Ni et al. (2015) observed that at least four Thr sites of MST2 (T349, T356, T364 and T378) are functional in MOB1 binding (see Figure 3D in Ni et al., 2015), but currently we only understand one out of these four residues on the structural level, namely the Thr378 site (see Figure 2 in Ni et al., 2015). Therefore, as already stressed above, it is quite possible that the positively charged Lys104/Lys105 residues of MOB1 (as defined in our study) may bond with the negatively charged phosphorylated Thr residues at 349, 356 and/or 364 on MST2 (or other negatively charged residues in the MST2 linker encompassing residues 321 to 370 or other regions of MST2/Hippo). We have expanded our main text on pages 17 and 18 to discuss these points.

Of course, we have also attempted to utilize the knowledge gained by the Ni et al. (2015) study in our settings. Therefore, based on Ni et al. (2015) we tested whether modifications of MOB1 residues involved in MST2 binding would offer a means to generate an alternative MOB1 mutant that

displays selective loss-of-interaction with human MST1/2 and fly Hippo, while binding to human LATS1/2, fly Warts, human NDR1/2 and fly Trc remain intact. In particular, we considered modifications of the MOB1 residues K153 and R154 as promising candidates due to their central roles in the phospho-threonine binding interface (see Figure 2 in Ni et al., 2015). Consequently, we generated and tested MOB1 K153A/R154A as well as K153E/R154E mutants (see new suppl. Figure S9). However as is apparent from our new suppl. Figure S9, MOB1(K153A/R154A) and MOB1(K153E/R154E) displayed defective binding to LATS2 and MST2, while NDR1 binding was intact. MOB1(K153A/R154A) and MOB1(K153E/R154E) were also defective in Warts and Hippo binding, while Trc binding was intact in insect cells (see new suppl. Figure S9).

Taken together, although the Ni et al. (2015) study did not experimentally test the complex formations of full-length wild-type MST2 with full-length MOB1 mutants, we studied full-length MOB1 mutants with mutations in the phospho-threonine binding interface. This revealed that residues K153 and R154 are important for MST2 as well as LATS2 binding, indicating that these residues are likely to represent a more general phosphor-serine/threonine binding domain as already suggested by the study of Rock et al. (2013). This conclusion is further supported by the finding that K153 and R154 of MOB1 contribute to Praja2 binding (see Figure 1d in Lignitto et al., 2013). Collectively, these data unfortunately rule out MOB1 K153/R154 mutants as alternative MOB1 variants with selective loss-of-interaction with MST1/2 and Hippo. (see also pages 11 and 17/18 of main text)

The authors of this manuscript produce their own Mob variant deficient in Mst2 binding by screening Mob1 variants with substitutions of charged, surface exposed amino acid residues in a low stringency co-immunoprecipitation assay. They identify two lysine residues (K104, and K105) as essential for stably binding to Mst1/2. The Mob variant they identify (MobK104EK105E) can however be efficiently phosphorylated in their kinase assay. They infer that Mst2 and MobK104EK105E interact transiently (via a low affinity interaction) and that that interaction is sufficient for their Mob variant to be phosphorylated.

RESPONSE:

Yes, based on a lack of MOB1/MST crystal structures when we started our project, we had to rely on the screening of MOB1 variants with specific substitutions. Furthermore, as already explained above, the now available MOB1/MST2 and MOB1/MST1 crystal structures (see Ni et al., 2015, and Couzens et al., 2017) did not enable us to generate a MOB1 variant with selective loss-of-interaction with human MST1/2 and fly Hippo. Based on our extensive biochemical characterization of our MOB1(K104E/K105E) mutant (see Figure 3 and suppl. Figures S5 to S13), we concluded that Lys104 and Lys105 of MOB1 are selectively required for binding to human MST1/2 and fly Hippo. Based on the observations that (i) kinase-substrate affinities can be weak, but still sufficient for efficient substrate phosphorylation (for example see Borders et al., 2002) and that (ii) transient kinase-substrate interactions can be below the detection limit of co-immunoprecipitation assays (for example see Piehler 2005), we were not surprised that MOB1(K104E/K105E) is still efficiently phosphorylated on Thr12 and Thr35 by human MST1/2 and fly Hippo (see suppl. Figures S12 and S13 in our revised manuscript). Most likely, human MST1/2 and fly Hippo can transiently interact with MOB1(K104E/K105E) to such a degree that MOB1(K104E/K105E) is sufficiently phosphorylated by its upstream kinases. We would not know whether this transient interaction is of low, medium or high affinity. The key point is anyways that all our assays (including also the newly added suppl. Figures S8 and S11) support the conclusion that MOB1(K104E/K105E) does not form a stable complex with MST1/2 and Hippo.

My concern is that the authors' variant retains more Mst2 binding activity than it appears in their assay, and therefore their in vivo conclusions are meaningless.

RESPONSE:

Of course, exactly this point has also been our main concern from the beginning. Therefore, we have performed various experiments to scrutinize the nature of our MOB1(K104E/K105E) mutant:

- (i) The characterization of MOB1(K104E/K105E) binding to human MST1/2 and fly Hippo was performed in low-stringency co-IP buffer (30mM HEPES pH7.4, 20mM beta-glycerophosphate, 20mM KCl, 1mM EGTA, 2mM NaF, 1mM Na₃VO₄, 1% TX-100 supplemented with protease and phosphatase inhibitors). We used our standard co-IP procedure that has worked very well and robustly in our hands for more than a decade. (see Figure 3 and suppl. Figures S5 to S7 in our revised manuscript version)
- (ii) The interaction of recombinant full-length MOB1(K104E/K105E) with recombinant MST2 (2-392) was also investigated using gel filtration chromatography in the following buffer system: 20 mM Tris pH8.0, 200 mM NaCl, and 5 mM DTT. (see suppl. Figure S10)
- (iii) The phosphorylation of recombinant full-length MOB1(K104E/K105E) by recombinant MST1/2 (or IP purified Hippo) was tested using standard kinase assay conditions. More specifically, we used buffer conditions that we are already successfully using since 2005 (and have been used by our kinase mentor, Dr. Brian A. Hemmings, for several decades). (see suppl. Figure S13)
- (iv) In addition we ensured that the available anti-phospho-T12 and anti-phospho-T35 antibodies are thoroughly characterized. (see suppl. Figure S12)
- (v) We are further including in the revised manuscript also new ITC assays in order to test the interaction of non-phospho vs. phosphorylated full-length MOB1(K104E/K105E) with the NTR domains of human NDR1, NDR2, LATS1 and LATS2. (Notably, these binding assays were performed in the following buffer system: 25 mM HEPES pH 7.5, 200 mM NaCl, and 1 mM EDTA.) (see new suppl. Figure S11)
- (vi) Last, but not least, we are including in the revised manuscript also novel co-IP experiments to further scrutinize the MOB1(K104E/K105E) variant. In addition to our low stringency co-IP buffer (see point (i) above), we performed co-IPs using the following three co-IP lysis buffers:
 -) RIPA buffer (20 mM Tris pH 7.5, 150 mM NaCl, 1 mM EDTA, 1 mM EGTA, 1% NP-40, 1% sodium deoxycholate, 2.5 mM sodium pyrophosphate, 1 mM β-glycerophosphate, 1 mM Na₃VO₄, supplemented with protease and phosphatase inhibitors)
 -) PBS-E lysis buffer (PBS, 1 mM EDTA, 1% (v/v) Triton X-100, 50 mM NaF, 1 mM Na₃VO₄, supplemented with protease and phosphatase inhibitors)
 -) 150mM NaCl lysis buffer (20 mM Tris pH 8.0, 150 mM NaCl, 10% glycerol, 1% NP-40, 5 mM EDTA, 0.5 mM EGTA, 20 mM β-glycerophosphate, 50 mM NaF, supplemented with protease and phosphatase inhibitors)(see new suppl. Figure S8)

In summary, using six different buffer systems and experimental approaches (see points (i), (ii), (v) and (vi) listed above) we detected that MOB1(K104E/K105E) does not form a stable (detectable) complex with MST1/2 (Hippo). Therefore, we are confidently concluding that MOB1(K104E/K105E) is deficient in stable MST1/2 (Hippo) binding.

The two accounts use different assays to assess binding and phosphorylation. Thus, at a minimum the authors should do a side by side comparison between the abilities of their variant and one of the Ni et al variants (MobKRR3A , for example) to bind Mst2 and be phosphorylated by Mst2. It is important that they use assays with appropriate dynamic range. For that, the Ni et al assays are good starting points as they are able to quantitate differences between the Mob variants.

RESPONSE:

As expected from two independent studies, Ni et al. (2015) and our study used different assays and approaches. The aim of our study was not to copy/repeat the study of Ni et al. (2015). The big goal of our study was to examine the biological significance of MOB1 interactions with Hippo core kinases, an important issue that was not covered by Ni et al. (2015).

As already described above, we have tested the binding of MOB1 K153/R154 mutants to MST2, LATS2 and NDR1 as well as fly Hippo, Warts and Trc (see new suppl. Figure S9). This revealed that the K153/R154 motif of MOB1 is required for MST2 (Hippo) as well as LATS2 (Warts) binding, indicating that these residues are likely to represent a more general phospho-serine/threonine binding domain as already suggested by the study of Rock et al. (2013). This conclusion is further supported by the finding that K153 and R154 of MOB1 are present in the domain central for Praja2 binding (see Lignitto et al., 2013). Collectively, these data unfortunately rule out MOB1 K153/R154 mutants as alternative MOB1 variants with selective loss-of-interaction with MST1/2 and Hippo. (Of note, MOB1 KRR3A corresponds to MOB1(K153A/R154A/R157A).)

Given the unexpected outcome of these additional experiments, we focused in our revised manuscript on expanding the characterization of our MOB1(K104E/K105E) as outlined above and below in our detailed responses.

Regarding the testing of MOB1 K153/R154 mutants as MST1/2 (Hippo) substrates, we do not think that this is necessary, since we already tested MOB1(E51K). MOB1(E51K) is defective in stable MST1/2, LATS1/2 and NDR1/2 binding (see Figure 3c for a summary), but efficiently phosphorylated on Thr12 and Thr35 by human MST1/2 and fly (see suppl. Figure S13). Thus, we have already tested a second MST1/2 binding deficient mutant of MOB1 as substrate of MST1/2.

We have updated the main text on pages 11, 12, 17 and 18 regarding these points.

Possible sources for the discrepancy between this manuscript and the Ni et al manuscript could include:

- *Differences in the ionic strength of the buffers used for co-immunoprecipitations between the two accounts. The low stringency buffer used by the manuscript under review is a buffer with sub physiological ionic strength; however that may not be low stringency for the Mob-Mst2 interaction, which could be driven by hydrophobic interactions.*

RESPONSE:

We fully agree with the reviewer that the ionic strength of the buffers must be taken into account when performing co-IPs and any other protein-protein interaction assay. Buffer systems usually contain one or more salts, with NaCl and/or KCl used as the most commonly used salts. Thus, we presented already in the initial manuscript version the characterization of MOB1(K104E/K105E) in two different buffer systems (one with 20mM KCl and detergent, and the other with 200m NaCl and no detergent) (see points (i) and (ii) below).

- (i) The characterization of MOB1(K104E/K105E) binding to human MST1/2 and fly Hippo was performed in low-stringency co-IP buffer (30mM HEPES pH7.4, 20mM beta-glycerophosphate, 20mM KCl, 1mM EGTA, 2mM NaF, 1mM Na3VO4, 1% TX-100) supplemented with protease and phosphatase inhibitors. We used our standard co-IP procedure that has worked very well and robustly in our hands for more than a decade. (see Figure 3 and suppl. Figures S5 to S7)

MST2 (2-392) was also investigated using gel filtration chromatography in the following buffer system: 20 mM Tris pH8.0, 200 mM NaCl, and 5 mM DTT.

(see suppl. Figure S10)

Collectively, based on these biochemical experiments we concluded in our initial manuscript version that MOB1(K104E/K105E) is deficient in MST1/2 (Hippo) binding. In order to further establish that MOB1(K104E/K105E) is deficient in MST1/2 (Hippo) binding, we now include the following novel experiments in our revised manuscript (see new suppl. Figures S8 and S11):

- a) Co-IP of MOB1(K104E/K105E) and MST2 in another standard lysis buffer (20 mM Tris pH 8.0, 150 mM NaCl, 10% glycerol, 1% NP-40, 5 mM EDTA, 0.5 mM EGTA, 20 mM β -glycerophosphate, 50 mM NaF, 1 mM Na_3VO_4 , 1 mM benzamidine, 4 μM leupeptin, 0.5 mM phenylmethylsulfonyl fluoride [PMSF], 1 μM microcystin, and 1 mM dithiothreitol).
- b) Co-IP of MOB1(K104E/K105E) and MST2 in RIPA lysis buffer (10 mM sodium phosphate pH 7.2, 150 mM NaCl, 1% NP-40, 0.1% (w/v) SDS, 1% (w/v) sodium deoxycholate, 2 mM EDTA, 50mM NaF, 0.2 mM Na_3VO_4 , 1 mM benzamidine, 4 μM leupeptin, 0.5 mM phenylmethylsulfonyl fluoride [PMSF], 1 μM microcystin, and 1 mM dithiothreitol).
- c) Co-IP of MOB1(K104E/K105E) and MST2 in PBS-E lysis buffer (PBS, 1 mM EDTA, 1% (v/v) Triton X-100, 50 mM NaF, 1 mM Na_3VO_4 , 1 mM benzamidine, 4 μM leupeptin, 0.5 mM phenylmethylsulfonyl fluoride [PMSF], 1 μM microcystin, and 1 mM dithiothreitol).
- d) ITC assays to investigate the interaction of non-phospho vs. phosphorylated full-length MOB1(K104E/K105E) with the NTR domains of human NDR1, NDR2, LATS1 and LATS2. (buffer system: 25 mM HEPES pH 7.5, 200 mM NaCl, and 1 mM EDTA.)

Noteworthy, all these additional experiments (see new suppl. Figures S8 and S11) fully support our conclusion that MOB1(K104E/K105E) is deficient in stable MST1/2 (Hippo) binding.

The main text has been expanded/adjusted on pages 11 and 12 of our revised manuscript in order to incorporate these new findings.

• Differences in the buffer and stoichiometry conditions of the kinase assay. The authors of this manuscript use no salt in their kinase buffer, and much higher relative amounts of kinase.

RESPONSE:

We have been using our kinase assay conditions very successfully since 2005. Noteworthy, these conditions have also been used by our kinase assay mentor, Dr. Brian A. Hemmings, for several decades. Therefore, we do not see the need to change our reliable and robust *in vitro* kinase assay set up. For our study the key point is anyways to determine the phosphorylation status of MOB1 variants in cells. Therefore, we are including in our revised manuscript the analyses of Thr12 and Thr35 phosphorylation of MOB1(K104E/K105E) stably expressed in MCF-7 cells (see new suppl. Figure S15). In agreement with our *in vitro* kinase assays (see suppl. Figures S12 and S13) this revealed that MOB1(K104E/K105E) is phosphorylated on Thr12 and Thr35 to a similar extent as observed for wild-type MOB1 (see new suppl. Figure S15).

Other experiments that would strengthen and broaden the scope of this account would be to test the Ni et al variants mentioned above (deficient in both Mst2 binding, and phosphorylation by Mst2) as well as the un-phosphorylatable Mob variants (Mob2TA, Ni et al) in the physiological assays developed by the current study.

RESPONSE:

We agree with the reviewer that if MOB1 K153/R154 mutants would represent variants with selective loss of interaction with MST1/2 (Hippo), it would be worth to test these mutants in our biological settings as described in Figures 4, 5 and 6. However, as we already stated above, MOB1 K153/R154 mutants are not selective for MST2 (Hippo), since they are also defective for LATS2 (Warts) binding (see new suppl. Figure S9). Therefore, we did not test MOB1 K153/R154 mutants in our biological settings. (see also the new text on pages 11 and 12 of our main text)

Regarding the testing of a MOB1(T12A/T35A) mutant in our assays, we do not understand how this would be informative in the context of MOB1/Hippo complex formation. It is already known for quite some time that MOB1(T12A/T35A) is impaired in LATS1/2 as well as NDR1/2 binding (see Praskova et al., 2008). We define herein that LATS1/2 (Warts) binding to MOB1 is important (based on the D63V mutant described in our manuscript). Maybe it would be interesting for a future study to understand what happens when you simultaneously block MOB1 binding to LATS1/2 (Warts) and NDR1/2 (Trc), but this is clearly out of the scope/focus of our current manuscript. In this regard, we are stressing on page 10 of our revised manuscript that "... the *in vivo* implications of MST1/2 (Hpo) phosphorylation of MOB1 need to be delineated in future studies".

In addition, the authors could devise a binding assay in a close to physiological setting. For example, in the Fig. 4 experiments, they show that both wild type Mob and the MobK104EK105E variant suppress MCF-7 cells proliferation. Can the Mob variant bind endogenous Mst2 in this situation where it is clearly functionally active?

RESPONSE:

As proposed, we attempted several times without any interruption to perform this additional experiment in MCF-7 cell extracts. However, we are very unfortunately facing a serious problem regarding the background immunoblot signals in our co-IP samples. Basically, the heavy chain antibody signal (and possibly also another protein source) make it impossible to detect a specific signal for MST2 (or MST1) in the 50-60kDa range. Therefore, we cannot draw any reliable conclusions on our obtained data sets. We have tried varying the amounts of input lysate, primary and secondary antibodies. We tried different primary antibodies and four different secondary antibodies (including light chain specific antibody as well as protein A/G coupled HRP), but without obtaining a specific MST2 (or MST1) signal. (We simply do not know why we are having these issues in MCF-7 cell extracts and it is impossible to predict how quickly this can be solved.)

Consequently, given this very disappointing endeavor with the MCF-7 extracts, we have very significantly expanded our in characterization of the MOB1(K104E/K105E) mutants (see our detailed responses to main point 2 above). We are confident that these additional experiments sufficiently strengthen our conclusion that MOB1(K104E/K105E) is defective in stable complex formation with MST1/2 (Hippo), while interactions with LATS1/2 (Warts) and NDR1/2 (Trc) remain intact.

Another concern about the MobK104EK105E is that it has increased affinity for Lats1 and that the increased affinity is sufficient to bypass an Mst2 binding requirement in in vivo settings.

RESPONSE:

To address this point, we performed ITC assays to determine the affinity of MOB1(K104E/K105E) for the NTR domains of human LATS1, LATS2, NDR1 and NDR2 and compared the obtained values

with MOB1(wt) (see new suppl. Figures S3 and S11 in the context of Figure 2 – please see also the new text on page 12 of the main text). Noteworthy, we compared the affinities of unphosphorylated and phospho-MOB1(K104E/K105E) with wild-type MOB1 in the context of wild-type and mutant NDRs of human NDR1, NDR2, LATS1 and LATS2. Collectively, this revealed that MOB1(K104E/K105E) bound with very similar affinities like wild-type MOB1 to all the NTRs that were tested. We also planned to probe the MOB1/LATS1 interaction in MCF-7 cell extracts. However, based on our difficulties regarding the detection of MST1/2 in our co-IP samples from MCF-7 cells (see previous point above), we could not complete this additional experiment.

Nevertheless, our new ITC assay data clearly support the notion that full-length MOB1(K104E/K105E) does not display altered affinities for NDR1, NDR2, LATS1 and LATS2 (at least in our *in vitro* conditions using the NTRs of these kinases as ligands) – see also point below.

Two recent papers (Ni et al, 2015 and Kim et al, 2016) show that Mobs have an auto-inhibited state, and that binding and/or phosphorylation by Mst2 relieves that inhibition by increasing Mob's affinity for NDR kinases. I am concerned that the MobK104EK105E used in this studies, as selectively abolishing Mob-Mst2 stable interactions, has altered affinity for NDR kinases, in particular an increased affinity that could compensate for the loss of Mst2 binding. To address this concern, the authors should compare quantitatively the interactions (p)Mob1 K104E K105E -Lats1/NDR2 and (p)Mob1 -Lats1/NDR2.

RESPONSE:

As suggested by the reviewer we significantly expanded on our ITC assays to measure the affinities of MOB1(K104E/K105E) to the wild-type NTR domains of human LATS1, LATS2, NDR1 and NDR2 (see new suppl. Figure S11). Importantly, as done for MOB1(wt), we compared the affinities of non-phosphorylated vs. phospho-MOB1(K104E/K105E). Significantly, we also determined the binding affinities of non-phospho vs. phospho-MOB1(K104E/K105E) to NDR1(Y31V), NDR2(Y32V), LATS1(V647Y) and LATS2(V610Y) (see new suppl. Figure S11). Collectively, these ITC assays revealed that full-length MOB1(K104E/K105E) has very similar binding affinities when compared to MOB1 wild-type as non-phospho- as well as phospho-protein (for all NTR domains of human LATS1, LATS2, NDR1 and NDR2).

Therefore, we are concluding that MOB1(K104E/K105E) does not display compensatory changes in its affinities to LATS1/2 and/or NDR1/2, at least as judged by ITC assays. The main text has been expanded on page 12 of our revised manuscript in order to incorporate these new data.

3. The “updated” NDR kinase activation model proposed in the Discussion is not consistent with available data and does not follow from the authors’ results.

*The authors should specifically discuss their contribution to the mechanistic picture of NDR kinase activation. Their results suggest that stable Mob binding to Lats1/2, but not to Mst1, is required for Lats1/2 activation in physiological settings. This argues against models that involve ternary complex formation (Ni et al, 2015), but as they are (not addressing the role of phosphorylation) they produce no new information towards understanding how Mob phosphorylation fits into the *in vivo*, physiological picture (which the authors depict as an established step in Fig. S12). The authors should acknowledge the difficulty of reconciling all data in a general model. Indeed Vrabioiu and Struhl results show that Hpo is not required for Mats binding to Warts and triggering an activating conformational change. Thus in the particular context of the *Drosophila* wing growth Hpo (and thus Hpo binding and Hpo phosphorylation) is not required for Mats-Lats1 binding, although Mats phosphorylation by a different kinase may be required. The authors’ model (Fig. S12) could exemplify a mechanism for a switch like activation of NDR kinases in specific biological contexts (for example, a situation when a growth response should be abruptly terminated) and can be presented while acknowledging its limitations.*

The authors should discuss the Ni et al mutants and suggest possible explanations for why their variant behaves differently than the Ni et al variants.

RESPONSE:

As suggested, we re-phrased/expanded the discussion on pages 19, 20 and 21 of our revised manuscript in order to include/discuss the points raised by the reviewer. Ni et al. mutants relevant to our study (i.e. K153/R154 modifications) are discussed on pages 11, 12, 17 and 18.

Minor points

• *Line 91 of the manuscript claims Mob1 variants with selective NDR1/2 impairment, but those variants were not described here (Mob1 E51K did not pass the criteria for selective impairment).*

RESPONSE:

On pages 4/5 of our revised manuscript the sentence at question has been corrected to:

“Thus, we characterized the interactions of Hippo core kinases with full-length MOB1 variants carrying specific point mutations, resulting in the discovery of MOB1 variants that are selectively impaired in their binding to MST1/2 (Hpo) or LATS1/2 (Wts) in human and fly cells.”

• *The statements in lines 128, 178-179, and 229-231 are superfluous and do not say much in the context of the specific results.*

RESPONSE:

As suggested, these three sentences have been entirely removed from the revised manuscript.

• *Line 472, what is assembly buffer?*

RESPONSE:

Assembly buffer is composed as follows: 20 mM Tris pH 8.0, 200 mM NaCl, and 5 mM DTT. The composition of the assembly buffer is described in the subsection “Protein expression and purification for structural and biochemical analyses” of the Methods section. (Crystals of the MOB1/NDR2 complex were obtained using assembly buffer – see Figures 1, 2 and suppl. Figure S2; the characterization of MOB1 variants using gel filtration chromatography was also performed in assembly buffer – see suppl. Figure S10).

Selected references (in particular relevant for our responses to reviewer #3)

- Borders, C. L., M. J. Snider, R. Wolfenden, and P. L. Edmiston. Determination of the affinity of each component of a quaternary transition state analogue complex of creatine kinase. *Biochemistry* 41:6995-7000 (2002).
- Couzens AL, Xiong S, Knight JD, Mao DY, Guettler S, Picaud S, Kurinov I, Filippakopoulos P, Sicheri F, Gingras AC. MOB1 mediated phospho-recognition in the core mammalian Hippo pathway. *Mol Cell Proteomics* doi: 10.1074/mcp.M116.065490. [Epub ahead of print] (2017).
- Kim SY, Tachioka Y, Mori T, Hakoshima T. Structural basis for autoinhibition and its relief of MOB1 in the Hippo pathway. *Sci Rep* 6, 28488 (2016).
- Lignitto L, Arcella A, Sepe M, Rinaldi L, Delle Donne R, Gallo A, Stefan E, Bachmann VA, Oliva MA, Tiziana Storlazzi C, L'Abbate A, Brunetti A, Gargiulo S, Gramanzini M, Insabato L, Garbi C, Gottesman ME, Feliciello A. Proteolysis of MOB1 by the ubiquitin ligase praja2 attenuates Hippo signalling and supports glioblastoma growth. *Nat Commun* 4:1822 (2013).
- Meng Z, Moroishi T, Guan KL. Mechanisms of Hippo pathway regulation. *Genes Dev.* 30(1):1-17 (2016).
- Ni L, Zheng Y, Hara M, Pan D, Luo X. Structural basis for Mob1-dependent activation of the core Mst-Lats kinase cascade in Hippo signaling. *Genes & development* 29, 1416-1431 (2015).
- Piehler, J. New methodologies for measuring protein interactions in vivo and in vitro. *Curr. Opin. Struct. Biol.* 15:4-14 (2005).
- Praskova M, Xia F, Avruch J. MOBKL1A/MOBKL1B phosphorylation by MST1 and MST2 inhibits cell proliferation. *Curr Biol.* 18(5):311-21 (2008).
- Rock JM, Lim D, Stach L, Odrodowicz RW, Keck JM, Jones MH, Wong CC, Yates JR 3rd, Winey M, Smerdon SJ, Yaffe MB, Amon A. Activation of the yeast Hippo pathway by phosphorylation-dependent assembly of signaling complexes. *Science* 17;340(6134):871-5 (2013).
- Vrabioiu AM, Struhl G. Fat/Dachsous Signaling Promotes Drosophila Wing Growth by Regulating the Conformational State of the NDR Kinase Warts. *Dev Cell* 35, 737-749 (2015).
- Xiong S, Couzens AL, Kean MJ, Mao DY, Guettler S, Kurinov I, Gingras AC, Sicheri F. Regulation of protein interactions by MOB1 phosphorylation. *Mol Cell Proteomics* doi: 10.1074/mcp.M117.068130. [Epub ahead of print] (2017).

Reviewers' Comments:

Reviewer #2:

Remarks to the Author:

The authors have addressed my main concerns and I am satisfied with their response.

Reviewer #3:

Remarks to the Author:

I looked over the revised manuscript and my initial comments. The authors have addressed my concerns with additional experiments, and text clarifications.

In particular, they discussed how their manuscript relates to the Ni et al., and two other recent reports (Couzens et al., 2017 and Xiong et al., 2017). I found their arguments convincing and I have a greater appreciation of their full length binding studies (the other reports use short NDR peptides). The additional experiments (testing of the Ni et al. Mob mutants in the context of full length NDR/Lats/Mst) further support their argument.

I am also pleased to see further characterization of the critical Mob K104E K105E mutant, in particular additional immunoprecipitation assays, and additional ITC experiments to determine the Mob mutant binding affinities for the NTRs of NDR1/2 and Lats1/2.

In addition to reagents, and mechanistic insights into the ongoing controversy about the activation of the core Hippo signaling cassette, the manuscript demonstrates the need to focus on in vivo, physiological studies. I think the manuscript is of great interest to the Hippo signaling field, and I thoroughly recommend publication in Nature Communications.